# Tsunamis boulders on the rocky shores of Minorca (Balearic Islands)

Francesc X. Roig-Munar[1], Joan M. Vilaplana[2], Antoni Rodríguez-Perea[3], José A. Martín-Prieto[1], Bernadí Gelabert[4]

1 QU4TRE, environmental consulting / AXIAL, geology and natural environment. Carritxaret 18.6, es Migjorn Gran, E07749 Minorca, Spain
2 Department of Earth and Ocean Dynamics, RISKNAT Group, Geomodels, Universitat de Barcelona. Martí i Franquès, s/n E08028 Barcelona. nue.vilaplana@ub.edu
Department of Geography, 4 Department of Biology, Universitat de les Illes Balears, Carretera de Valldemossa, km 7.5, E07122, Palma, Majorca, Spain

*Corresponding author:* Bernadí Gelabert (bernadi.gelabert@uib.es)

**Abstract** Large boulders have been found on marine cliffs of 24 study areas on Minorca, in the Balearic Archipelago. These large imbricated boulders, of up to 229 tonnes, are located on platforms that conform the rocky coastline of Minorca, several tenths of meters from the edge of the cliff, up to 15 m above the sea level, and kilometres away from any inland escarpment. They are mostly located on the southeast coast of the island, and numerical models have identified this coastline as a zone with a high probability of tsunami impact. The age of the boulders of the studied localities range between 1574 AD and recent time, although most of them are concentrated around the year 1790 AD. Although some storm waves might have play a role in their dislodging, the distribution of the boulder sites along the Balearic Islands, the degree and direction of imbrication and the run-up necessary for their placement, suggest a transport from North African tsunami waves that hit the coastline of Minorca.

## 1 Introduction

Although they are less frequent than those of the Pacific and Indian oceans, tsunamis in the Mediterranean Sea are well known from historical accounts (Soloviev, 1990). Large boulder accumulations observed and studied on various coastlines of the Western Mediterranean have been associated with extreme wave events (tsunamis or storms): France (Shah-Hosseini et al. 2013), Southern Italy (Barbano et al. 2010, 2011; Mastronuzzi et al. 2007; Mastronuzzi and Pignatelli 2012; Pignatelli et al.

2009; Scicchitano et al. 2007, 2012), and Algeria (Maouche et al. 2009). Large boulders placed over coastal rocky cliffs on Minorca Island have been found mainly on the southeast and west coastline (Roig-Munar, 2016) (Fig. 1). Some are positioned well above the maximum stand of any recorded storm wave (up to 27 m), many show imbricated boulder ridges, and all of them are located away from any high inland relief that might explain an origin from gravitational fall.

The presence of large boulders on the rocky shores of the Balearic Islands has been treated by Bartel and Kelletat (2003), Schefers and Kelletat (2003) and Kelletat et al. (2005), but only on the island of Majorca. The authors linked the presence of large boulders on the coastal platform of Majorca with storm waves and/or tsunami processes, establishing a simple equation (Transport Figure) to discern those displaced by a storm wave or a tsunami event. In fact, in many areas of the Western Mediterranean, imbricated, metric size boulders have been interpreted as remnants of the tsunamis occurred in the last centuries 10 (Pignatelly et al., 2009). Only at the Atlantic coast, with much higher fetch, storm wave period and tidal range, imbricated boulders at high altitudes are tied to storm processes (Hanson and Hall, 2009; Etienne and Paris 2010; Hall, 2011). However, the distinction between tsunami or storm boulders is not easy nor without controversy, though it is based on a set of sedimentological, morphological and chronological criteria to be treated in each case (Scheffers and Kinis, 2014). The main goal of this article is to demonstrate that some of the boulders located close to the coastal cliffs of Minorca were transported 15 and deposited by tsunamis that occurred in the recent past and mostly originated from submarine earthquakes at the Algerian coast.

## 2. Study site

### 2.1. Geology of the study areas

Both from a geological and geomorphological point of view, Minorca is divided into two parts separated by an imaginary line WNW-ESE that extends from Maó to Cala Morell (Fig. 1): a) the Migjorn, which covers the southern half of Minorca, is formed by undeformed calcareous materials from the upper Miocene forming a nearly horizontal platform; and b) the Tramuntana, which includes all the outcrops of Palaeozoic, Mesozoic and Oligocene age. These materials are faulted and folded by the alpine orogeny and constitute the northern half of the island characterized by gentle hills and valleys.

The eight study sites of the southern sector (Figure 1) and the eight study sites of the western sector are located on carbonated, horizontal, well-developed bedding, Upper Miocene rocks forming a marine cliff with heights between 4.5 and 20 m. On the other hand, five of the eight study sites of the northern sector corresponds to outcrops of massive Jurassic limestones, forming sea-cliffs between 2 and 20 m height. The other three study sites of the northern area are located on Plioquaternary eolianites: Tirant and Tusqueta sites constituting a gentle ramp where cliffs are absent; nevertheless, in Punta Grossa, eolianites conform 30 an 8 m high coastal cliff.

### 2.2. Maritime climate

The Mediterranean basin is characterized by a highly indented coastline that creates some small and well-defined sub-basins, where wave energy is conditioned by wind speed and by limited fetches (Lionello et al., 2006). In the western Mediterranean, the most intense waves come from the NE (Sotillo et al., 2005), although the NW also generates strong waves between the Balearics, Corsica and Sardinia (Bertotti and Cavaleri, 2008).

The coast of Minorca island is subject to a maritime climate characterized in the last 50 years by a maximum wave height of 10 m from a NNE dominant direction (Cañellas, 2010) (Fig. 1). The eastern coast of the island is characterized by a maximum wave height of 8.5 m with a dominant N component (Cañellas, 2010). At the northern sector of the Island, the maximum wave height recorded since 1958 was 11 m height from a NNE direction. The Hs50 is estimated at 9.88 m (Cañellas, 2010). Monthly maximum periods calculated for WANNA points around Minorca are between 11 and 14 seconds (Fig. 1). The tidal regime in

Minorca is of very low amplitude (30 cm), almost negligible for this study.

Mediterranean hurricanes, called medicanes in the Mediterranean, generated by intense tropical cyclones may be a more likely extreme waveform reaching the coast of Minorca. The remarkable the medicane of 10-11 November 2001 was associated with the seventh most intense cyclone around the Mediterranean, throughout the period ERA-40 (1957-2002) and is the most intense of all detected in the westernmost Mediterranean, near the Balearic Islands (Genovese et al., 2006). The wind exceeded 150

km/h, affecting a large marine extension and causing waves up to eleven meters of significant height (Jansà, 2003). The number of intense cyclones affecting the Balearic Islands during the period 1957-2007 is between 5 and 10 (Homar et al., 2007).

According to Papadopoulos (2009), the major tsunamigenic source in the Western Mediterranean is located north of Algeria (Fig. 2). The last tsunami registered was in the 2003 and produced great damages in several marinas and entrances of the Balearic Islands. Roger and Herber (2008) made numerical simulation of this tsunami affecting the Balearic Islands (Fig. 3).

Several seismic tsunamis have been recorded in the Balearic Islands; some of them have been described in chronicles as Fontseré, 1918 (Table 1).

**3 Methodology**

In this study, 3.144 boulders located in 24 areas of Minorca Island (Fig. 1) have been analysed. Boulder size was measured, as well as height above sea level, and the distance from the edge of the cliff. Orientation and imbrication were also considered,

together with their geomorphological context (Fig. 4). Transport Figure TF (Scheffers and Kelletat, 2003) was used to assess the power needed to dislodge and transport each boulder. TF is calculated as the product of the height above sea level, distance from the edge of the cliff, and weight. Scheffers and Kelletat (2003) consider boulders with TF>250 as indicative of tsunami boulders. In this paper we focus our study on boulders with TF>1000 and on boulders found on cliffs well above the maximum storm wave height recorded in Minorca, which is 11 m (Cañellas, 2010).

Calculation of boulder weights requires a good estimation of density and volume (Engel and May, 2012). In most cases the product of the three axis -a (length), b (width) and c (height) - of each boulder exceeds the true volume of the boulder. Sampling comparisons have been made between Vabc, and a more precise measurement obtained by triangulating the boulder in

homogeneous parallelepipeds (Fig. 5a). This procedure produced a correction coefficient of 0.62 that has been applied to all boulders analysed in this study. Densities of each lithology were calculated using the Archimedean principle of buoyancy in sea water.

In addition to TF, different equations (Table 2) have been applied to all the localities to calculate height of water required to dislodge and/or move each boulder. Nott (2003) has defined pre-settings for transported boulders (submerged, subaerial and joint bounded boulders JBB), and for each boulder type, a different equation for both tsunami and storm waves. Most of Minorcan boulders were dislodged from cliff edges (Fig. 6), so joint bounded and subaerial scenarios must be considered. Only nine boulders show features (marine fauna or notch fragments) defining they were originally submerged. Pignatelli (2009) defined a new equation to obtain the minimum tsunami height HT that can move a joint bounded boulder (JBB). The Nott derived equation differs from the original in the relevance of the c-axis that indicates the thickness of the boulder directly exposed to the wave impact. Engel and May (2012) reconsider Nott's equations using more accurate volume and density measurements, and defining equations to derive the minimum wave height of a tsunami HT or storm wave HS, that is required to dislodge a submerged, subaerial or JBB boulder (Table 2).

Age of the boulders was determined using two different methods: a) radiocarbon dating of marine incrusting fauna, and b) dating surface post-transport features. Most of the boulders show unconformable post-depositional solution pans on the surface, related to karstic dissolutions after the transport of the boulder. Some (Fig. 5b) of these post-depositional solution pans are intersecting pre-existing ones developed conformably with stratification. Karstic dissolution rate of these pans was estimated at average of 0.3 mm/y (Emery, 1946. Gómez-Pujol et al, 2002). Transport age of 145 boulders from 12 locations was determined using these two methods (Fig. 10).

Other qualitative observations were taken into account: a) relation of the boulders with their source area and presence of fractures that can promote detachment of the boulders, b) presence of incrusting of boring marine fauna indicating the origin of the boulder before its displacement, c) presence of pre-detachment and post-detachment solution pans which have been used as date indicators of boulder emplacement, d) degree of rounding of the boulders, presence or absence of other type of sediment as well as presence of abrasion surfaces due to boulder quarrying and transport and, e) presence of "flowouts" which are areas with denudated beds forming channels over the cliff favouring the entry and acceleration of the water flows and leaving a boulder ridge in its front.

## 4. Results

The 24 areas analysed (Figure 1) have been grouped into three sectors: SE, W and N. All the boulders were processed, but those with a TF lower than 1000 were excluded from the final analysis. Therefore, results are based on the analysis of 720 boulders.

### 4.1. Southeast sector

Although 1.766 boulders have been analysed in eight areas of the SE sector (Fig. 1 and 7), only 274 (16%) had a TF>1000. These boulders have an average size of 3.1 m along their longest axis (a), 2.16 m along the intermediate axis (b) and 0.9 m along the shortest axis (c), which usually corresponds to the thickness of the source strata. Mean weight is 11.62 t, with a maximum of 229 t on the coastal islet of Illa de l'Aire. Average cliff height is 6.8 m, average height of the boulders is 7.19 m,

and average distance from the edge of the cliff is 61.4 m, with extremes of 18.5 m and 136 m respectively. The highest regional storm wave registered was 7.5 m (Cañellas, 2010).

Engel and May (2012) formulations show that the boulders with a TF> 1000 from this sector require a column of water between 8.8 m (subaerial) and 14.4 m (JBB) to explain storm wave run-ups, and between 7.3 and 8.7 m for the tsunami run-ups.

We calculated that 33 % of the TF>1000 boulders are in areas above the maximum stand of the waves registered (7.5 m), and

many of them show imbrication patterns. Due to these two reasons, we interpreted these boulder deposits as produced by tsunami events. However, 79 % of all the boulders are positioned at a height at which they can be reworked by storm waves. Boulder setting of this sector can be characterized by the presence of several ridges of imbricate boulders (five of the eight sites show this setting) (Fig. 7), as well as sub-rounded boulders (5 of 8), and isolate groups of imbricate boulders (4 of 8). Although cliff altitude of this sector is quite low (6.8 m, average), and many sites show sub-rounded blocks (5 of 8), there is

not any clear relationship between these characters. As an example, some of the lower cliffs do not show any ridge; meanwhile some with higher cliffs do have ridges.

### 4.2. Western Sector

Along the cliffs of the western area (Fig. 1 and 8) 1.043 boulders were measured, and 232 boulders (22%) showed a TF>1000. These boulders have an average size of 2.38 m along the longest axis (a), 1.86 m along the intermediate axis (b) and 0.68 m

along the shortest axis (c), which mostly corresponds to the thickness of the source strata. Mean weight of these boulders is 4.6 t, with a maximum of 21.9 t. Average cliff height is 12 m, and the average boulder height is 16 m and at a distance of 40 m from the edge of the cliff, with extremes of 31 m and 65 m. The highest regional wave registered was 8 m (Cañellas, 2010). Formulations of Engel and May (2012) show that the boulders with a TF> 1000 require a column of water between 13.7 m (subaerial) and 18.6 m (JBB) to explain storm wave run-ups, and between 12.4 and 13.6 m for the tsunami run-ups. Almost all

the TF>1000 boulders are positioned above the maximum stand for waves registered along the west coast of Minorca (8 m). Only 16 % of all the boulders are positioned at a height at which they can be reworked by storm waves. The storm run-up heights for these boulders are out of the reach of storm waves.

Boulder setting of the Western sector of Minorca is characterized by higher cliff altitudes and imbricate boulder ridges at half of the sites analysed (4 of 8). Only two of the sites show sub-rounded boulders –the lower sites– and just one has isolated

groups of imbricate boulders.

### 4.3. Northern sector

Along the North coast of Minorca 338 boulders have been measured (Fig. 1 and 9), and 214 (63%) showed a TF>1000. The boulders have an average size of 2.56 m along longest axis (a), 1.94 m along the intermediate axis (b) and 1.3 m along the shortest axis (c). Mean weight of these boulders is 12.07 t, with a maximum of 128.3 t at Illa dels Porros. Average cliff height is 7.81 m; the average boulder height is 11.7 m and at a distance of 66.2 m from the edge of the cliff, with extremes of 27 m and 129 m. The highest regional wave height was calculated at 11 m (Cañelles, 2010).

Formulations of Engel and May (2012) show that the boulders with TF> 1000 require a column of water between 9.8 m (subaerial) and 21.6 m (JBB) to explain storm wave run-ups, and between 8.3 and 11.3 m for the tsunami run-ups. Most of the TF>1000 boulders (74%) are positioned above the maximum wave height registered along the North coast of Minorca (9 m). In addition, 24 % of the boulders are positioned at a height at which they can be reworked by storm waves. The storm run-up heights for these boulders of this sector are out of the reach of storm waves.

Few imbricate ridges (just two of the eight sites), only one site with isolated imbricate groups of boulders, and a greater presence of sub-rounded blocks (6 of 8) characterize the setting of the Northern boulders.

### 4.4. Biggest boulders

The results for each area indicate the average size and weight for all the boulders with a TF>1000, but we will consider some our findings about the largest boulders of each area. The largest boulders of the SE area of Minorca are located on Illa de l'Aire (Fig. 7), just 960 m off the SE coastal tip of Minorca. The largest boulders of this area weigh 228 t, 154 t and 114 t. Engel and May (2012) equations provide storm run-ups estimations of 32 m, 23 m and 22 m respectively, meanwhile for a tsunami run-up they required 12 m, 9 m and 9 m.

The largest boulders of the Western area of Minorca weigh 21.9 t, 18.2 t and 16.8 t, but they are located higher and more inland than those of the SE coast. The results of Engel and May (2012) equations of this area show storm run-ups of 20.2 m, 16.4 m and 16.5 m and tsunami run-ups of 9.9 m, 10.5 m and 10.5 m.

The North coast largest boulders weigh 128.3 t, 56.5 t and 53.7 t. They are found on the small islet of Illa des Porros (Fig. 9), just 426 m off the Northern tip of Minorca (Fig. 9). According to the equations of Engel and May (2012), storm run-ups of 46.3 m, 45.4 m and 37.7 m are required to transport these boulders, and heights of 19.8 m, 22.6 m and 16.6 m for a tsunami run-up.

### 4.5. Dating Age of the deposits

Five of the analysed boulders show marine fauna, indicating that they have been dislodged from the submerged area and deposited above the cliff. Two of these boulders have been sampled for 14C dating: A boulder from Son Ganxo (SE of Minorca, Fig. 7) is a fragment of shoreline notch (wave-cut notch); located 2.5 m above sea level, at a distance of 18.4 m from the cliff edge, with a weight of 4.75 tons. Radiocarbon dating determined an age younger than 1964 AD (RICH-21441: 106.96 ± 0.39 BP, calibrated after 1965 AD with the marine curve). Another boulder in Sant Esteve (SE of Minorca, Fig. 7) is situated about

19 meters from the waterfront and 1 m above sea level, with a weight of 43.15 t, and 14C dating determined an age younger than 1720 AD (RICH-21442: 518 $\pm$ 31 BP, cal AD 1720-1950 for 95.4% and cal AD 1804-1910 for 68.3 %).

Some of the boulders in the spray areas show post-depositional dissolution pans (Fig. 5b). Although dissolution rate for these pans is not uniform (it increases near the cliff edge), we have considered an average of 0.3 mm/y (Emery, 1946. Gómez-Pujol et al, 2002). This rate has been used to date the age of 145 pans found on the surface of the boulders (Fig. 10).

Radiocarbon dating and estimating dates using dissolution ratios, provided a range of ages for 12 locations between 1574 and 1813 AD, although 8 of the 12 dates are situated around the year 1790 AD (Fig. 10).

These results situate the processes that lead to the deposition of blocks in a few hundred years, discarding geologically older events. In all likelihood, there were previous events that either were obliterated by the youngest and most intense or have not yet been possible to identify.

## 5. Discussion

In interpreting the cause of extreme wave events, there are two feasible hypotheses, namely tsunami waves or storm waves. The formers are long period waves (up to $10^2$ minutes) of long wavelength (>100 km), the latter characterised by much shorter period (max. 25 secs) and length ($10^2$ m). Because of their long wavelength, tsunami waves possess a minimum factor of 4x greater power in relation to their height than storm waves (Mottershead et al. 2014). The impact of a tsunami on a cliff has to be compared to that of a flood, since the mass of water, overcoming the edge of the cliff, produces a flow inland equivalent to a massive flooding. On the other hand, the action of the storm waves, as well as being more local, more depending on the conditions predicted by the cliff (fractures, abrasion caves, etc.), depends a lot on the bathymetry prior to the cliff, which determines the slope of the wave and the distance of its break. In the Balearic Islands, the comparison between the run-ups of tsunamis and storm waves must also consider their proximity to tsunamigenic sources and the reduced fetch available for the storm waves, especially those that come from the South. This greater power enables tsunami to achieve both detachment of significantly larger bedrock clasts and also much greater run-up heights and run-in distances.

Small recent tsunamis have affected the island of Minorca as stated by local newspapers (Diario de Menorca, 2003, 22nd and 23rd may). The tsunamigenic source is the Algerian coast, which according to the historical and instrumental seismicity is exposed to relevant seismic hazards and risks (Papadopoulos, 2009). The last tsunami that affected Minorca Island was generated by the Zemmouri (Algeria) earthquake that took place on May 21, 2003, with a magnitude of 6.9 Mw. This earthquake was generated by a reverse fault, leading to a significant deformation of the seabed, and creating a tsunami that was observed in Algeria and Spain, and even reached the coasts of France and Italy. This event leaded 3 m high waves in Ibiza, the highest tsunami waves recorded in recent years in the Balearic Islands, which damaged some of the harbour facilities on Minorca, Majorca and Ibiza. A fragment of the chronicle about the tsunami in Diario de Menorca (22/05/2003) stated: "In the case of the Port of Maó (the capital city of Minorca), the movement of the waters was spectacular: no sooner had it disappeared from the shore, leaving the bottom of the harbour uncovered, then it returned, flooding the seafront and even the road. The

same situation was experienced simultaneously in Cales Fonts, Cala Alcaufar and Cala Sant Esteve (three calas in the E coast of Minorca), where some hammocks were 300 m from the beach, along with dead fish" (see figure 7 for location). Unfortunately, we did not study the effects of the tsunami on the boulders at that time. Tsunami simulations of this event (Fig. 3) were performed by several authors (Hébert and Alasset, 2003; Alasset et al., 2006, Roger and Hebert, 2008).

Thus, there is currently seismic activity at the bottom of the Algerian Basin that gives rise to tsunamis affecting the coast of Minorca. In the recent past, in the last 500 years, there have been tsunamis affecting the Balearic Islands (Table 1). There are also historical tsunami records reporting a flooding event with a run-in of to 2 km inland in Santanyí (location on Fig. 2), on the east coast of Majorca in 1756 (Fontsere, 1918). Numerical models of tsunami simulation from submarine earthquakes at the North African Coast (i.e. Alvarez et al., 2011; Roger and Hebert, 2008) show that the southeast and west of Minorca would

be one of the most affected areas by the tsunami impacts. On the contrary, the fetch length for the southern coast of Minorca is relatively low: 300 km in the S direction and 500 km in the E direction. Thus, in the last 50 years the maximum extremal wave height detected in an offshore buoy was of 11 m high at the 2001 medicane (Jansà, 2013).

According to Papadopoulos (2009), the major tsunamigenic source in the Western Mediterranean is located north of Algeria (Figure 2), although the Alborán region has to be taken into account too. In other areas as the Liguro-Provençal basin and the

Valencia Trough (Fig. 2), the seismicity is too low to be taken into account as tsunamigenic areas. The seismicity of the northern region of Algeria is dominated by thrust focal mechanisms to the west and central part of this area and by strike-slip faults to the east (e.g., Bezzeghoud et al., 2014). The Alboran region is dominated by strike-slip and extensional focal mechanisms where the largest magnitudes are usually low to moderate (Vanucci et al., 2004).

If we focus in North Algeria, since 1716, there have been 7 seismic events (Fig. 10) with intensity greater than X recorded by

Ayadi and Besseghoud (2014) capable of originating a tsunami that, according to the numerical models, will directly hit the coast of Minorca (especially the southern one). According to the same authors, only one seismic event of high intensity is recorded prior to 1716: Algiers, third of January of 1365. Thus, between the period 1716-2017 seven high magnitude events have been recorded, whereas between 1365 and 1715 only one high magnitude event has been recorded. This fact is probably due to the lack of information as we go back in time and probably the frequency of the first period must be hidden in some

way.

The geographical distribution of boulders sites (Figs. 1 and 3) in the Balearic Islands gives clear indications of their tsunamitic origin. Boulders sites in Majorca are distributed along the eastern and southern coast and the same happens in Ibiza. Only in Minorca we found boulder sites at the north coast, despite most of the boulder settings are located in the south and coast of the island. In figure 3 we show the perfect correspondence between the expected locations where a northern Africa generated

tsunami will hit the Balearic Islands (from numerical model simulation) and the sites where boulder accumulations are. Storm waves have larger fetch in the northern coast of the Balearic Islands, but almost no large boulders have been found at the western and northern coast of Majorca, neither at the northern coast of Ibiza.

Despite we are aware that hydrodynamic equations need review (Cox et al., 2018) and they are not a definitive approach for discerning tsunami from storm boulders, we used Engels and May, Nott and Pignatelly equations. The Engel and May equation

calculates the wave height needed to transport boulders located at sea level. The height at which the boulders are is not contemplated in this equation. In Table 3 we present the average results for joint bounded boulders (JBB) of the three sectors studied. Hs (storm wave height needed to move a boulder at sea level) is, in the three sectors, approximately four times Ht (tsunami wave height needed to move a boulder at sea level). Because Hs is approximately equal (or higher) to Hm (the

maximum wave height recorded), storms of Minorca cannot move any boulder with Transport Figure>1000, not even the ones located at sea level! This point agrees with our observations in the field that the biggest storm wave ever recorded in Minorca moved none of the boulders we mark in advance.  In the other hand, tsunami waves heights of less than 4 m at the northern sector or less than 2 m in the western and southern sectors can move boulders with TF>1000 at sea level. The run-up values given in the text are the sum of Hs and Alt (the average altitude of boulders), for storms run-up (Rs) and the sum of Ht and Alt

for tsunami run-up (Rt).  Along the SE sector of Minorca coastline, for joint bounded boulders, storm run-ups of 14.4 m are required to explain the position of the boulders, while only 8.7 m tsunami run-ups can explain the same positions. Results along the higher cliffs of the W coastline, requires tsunamis run-ups 13.6 m high and/or storm run-ups of 18.6 m. The calculations along the northern coast sector require storm run-ups of more than 21 m, which are not plausible, while the height of a tsunami run-up required to position the boulders is 11.3 meters.

According the setting of the boulders and the results of the hydrodynamic equations, it seems clear than large boulders cannot be transported by a single storm event, neither by a series of storms. On the other hand, hydrodynamic equations require run-ups of the tsunami wave that multiply, between two and ten times, the heights that models forecast for tsunami waves in the open sea. First of all, the run-up of tsunamis on vertical cliffs is several times higher than that occurring on low coastal areas (Bryant, 2014). Run-up is also enhanced due to several factors (Lekkas et al., 2011): 1) by the distance from the tsunami

generation area (of only 300 km in our case), 2) by the narrowness of the continental shelf (as in Minorca), 3) by the fact than the tsunami propagation vector is almost perpendicular to the main shoreline direction, and 4) by land morphology, characterized by vertical cliffs with entrances (calas). For these reasons, run-ups heights on Minorca would have been several times higher than tsunami wave heights. On the contrary, as they shoal, wave heights increase run-up heights in a much lesser way and thus, it is impossible to reach the run-up values obtained from the hydrodynamic equations.

Recent examples in the Balearic Islands confirm the last statement: the tsunami of 2003 had an offshore wave height of 30-40 cm (according to simulations) and reach the western part of Ibiza with a run-up of 3 m, which means a multiplying factor of x10.  In the other hand, in November of 2017, a severe storm caused waves of up to 11 m offshore north of Minorca. These waves, after breaking, decreased their height when arriving at the coast of Minorca. A field survey, made days after the storm, reveal that none the boulders marked in advance (even those located at only 1 m above sea level) moved, neither new blocks

appeared.

Regarding the dating of the boulders, although only two blocks with embedded marine fauna (and located only 1 m above the sea level) have been radiocarbon dated, such dates serve as a reference to the second dating method used. Our C14 results show than in one case a block was moved after 1856 AD, and in the other case was transported after 1964.

The second dating method used is based on an average dissolution rate of dissolution pans. This requires identifying post-depositional dissolution pans, that is, those that have been formed after the movement of the boulders. They can be formed on the same boulder once transported or on the denudation surface that results from the quarry of the boulder. A margin of error can be established based on the variability of the dissolution rate, which is not very high because the boulders are located away from the cliff edge, where the dissolution rate is more variable. However, in no way do the resulting values (age values) match with marine levels different from the current one. Other similar boulders dated by Kelletat (2005) on the neighbouring island of Majorca, correspond to ages between 565 AD and 1508 AD.

Estimations using dissolution rates of surface pans are coherent with the two macro-fauna radiocarbon C14 dates. Historic records of earthquakes and associated tsunamis (Fontseré, 1918; Martinez-Solares, 2001; Silva and Rodriguez, 2014) are also consistent with our chronology (Figure 10). Among the historical records of huge wave phenomena that have affected the Balearic Islands, there are also some episodes that can be attributed to tsunamis. In 1856, the chronicles written by Fontseré (1918) record an extraordinary sea rise in the Port of Maó (Minorca) that destroys several moorings. In 1918, a new 'seismic wave' floods the Port of Maó, following an earthquake offshore of the Algerian coast (Fontseré, 1918). The data of the National Geographic Institute of Spain (Martinez-Solares, 2001 and Silva and Rodríguez, 2014) record in 1756 the presence of a tsunami that flooded more than 2.4 km inland in Santanyí (location on Fig.2), at the southern coast of Majorca (Fontseré, 1918). A run-up up to 45 m must be deduced from that description. In all likelihood, some tsunamis have not been reflected in the historical chronicles because in the recent past (18th, 19th and early 20th century's) the coastal part of the Balearic Islands were uninhabited. Only the tsunamis that historically affected the towns near the coast were perceived.

Finally, settings of the boulders depend on local physiography and on the characteristics of the flow that transported them. Most of the imbricate ridges are found along the SE sector, with lower cliffs and a bigger impact of potential tsunamis. Up to 62 % of the boulders along the SE coastline are sub-rounded, indicating some reworking by storm waves. Boulders along the western sites are positioned higher, and only 25% are sub-rounded, overlapping with the presence of flow-out morphologies. Most of the boulders of this sector have been detached and transported by tsunami flows, but storm waves have moved some boulders several centimetres, reworking them locally. The position of the boulders along the North coast sector shows evidences of both tsunami, and storm wave flows: 75 % of the sites have sub-rounded blocks and just 25 % of the sites have imbricate ridges. Weight, distance inland and height of some boulders, cannot be explained by storm waves. The tsunamis hitting the north coast of Minorca could be caused by a refraction of a tsunami wave originated off the North Africa coast but we do not exclude submarine landslides occurring off the Catalan platform or at the Liguro-Provençal basin platform (Fig. 3).

**Conclusions**

More than three thousand large boulders have been analysed on the coastal platforms of Minorca, of which 720 (the ones with larger Transport Figure values) have been selected for this study. Weight, height above sea level and distance from the edge

of the cliff, indicate that they have been dislodged and positioned by the action of tsunami waves, although some of these boulders have also been reworked by storm waves.

Boulder sites in the Balearic Islands are mainly located in the southern and eastern parts of the islands. This fact is decisive to demonstrate that they have been transported by tsunamis and not by storms: whereas the prevailing and strongest wind comes from the north, the main tsunamigenic area is the Algerian coast, located S-SE of the Balearic Islands.

Tsunamis generated off the Algerian coast are quite well known. What was little known is the potential impact of these waves on the coastline of the Balearic Islands, including Minorca. Tsunami simulation models have confirmed the high probability of tsunami wave impact along the coast of the Balearic Islands. The historical chronicles of tsunami events hitting the Islands have supported these models. The last 2003 tsunami episode caused important damages in some harbours of the Balearic Islands.

Despite the location of the boulders being a very important issue, further information obtained from boulder orientations and the presence imbricated ridges and/or isolated groups of imbricated boulders, are evidences of a continuous flow which can only be originated by a tsunami. Distance from local escarpments can exclude that any of the boulders analysed had its origin from a rock fall.

Hydrodynamic equations applied to these boulders give wave run-up values that are very far from the reach of the waves recorded in the last 50 years, a clear indication that a tsunami wave was the cause of their dislodgement, transport and setting. Weights up to 228 t (Illa de l'Aire, Fig. 7), altitudes reaching 31 m (Punta Nati, Fig. 8) above sea level, and distances from the cliff edge of up to 136 m (Illa de l'Aire), confirm the results obtained in our calculations. Historic data of storm waves, or even medicane (11 m) events, cannot explain the size and positioning of the boulders.

Dating by 14C and obtained from pan dissolution rates establish an age range for tsunami emplacement of the studied boulders between the 17th and 19th centuries. During this period, seven earthquakes with intensities larger than X have been documented along the North Algerian coast and 11 historical tsunami phenomena have been described from historical records in the Balearic Islands.

Acknowledgments

This study was supported by the CHARMA Project (MINECO, Ref.: CGL2013-40828-R), the CGL2013-48441-P, the CGL2016-79246-P (AEI/FEDER, UE) and the CSO20015-64468-P (MINECO/FEDER) projects.

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

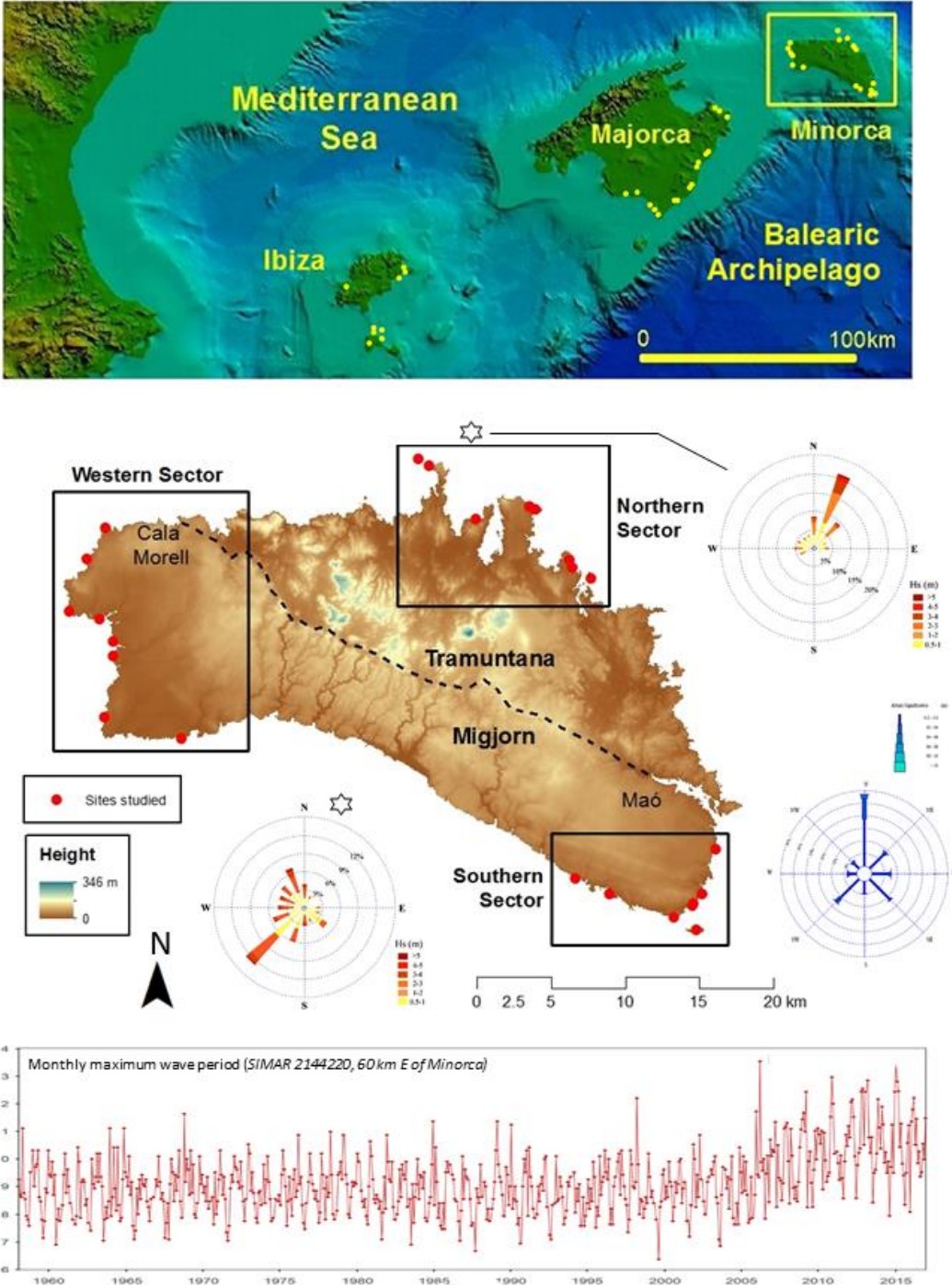

**Figure 1: Up)** Boulder sites at the Balearic Islands. **Middle)** Situation of the sampled areas: A) West, B) North and C) Southeast of Minorca and some rose diagrams of waves (orange) and wind regime (blue). **Down)** Monthly maximum wave period (in seconds) representative of Minorca wave regime.

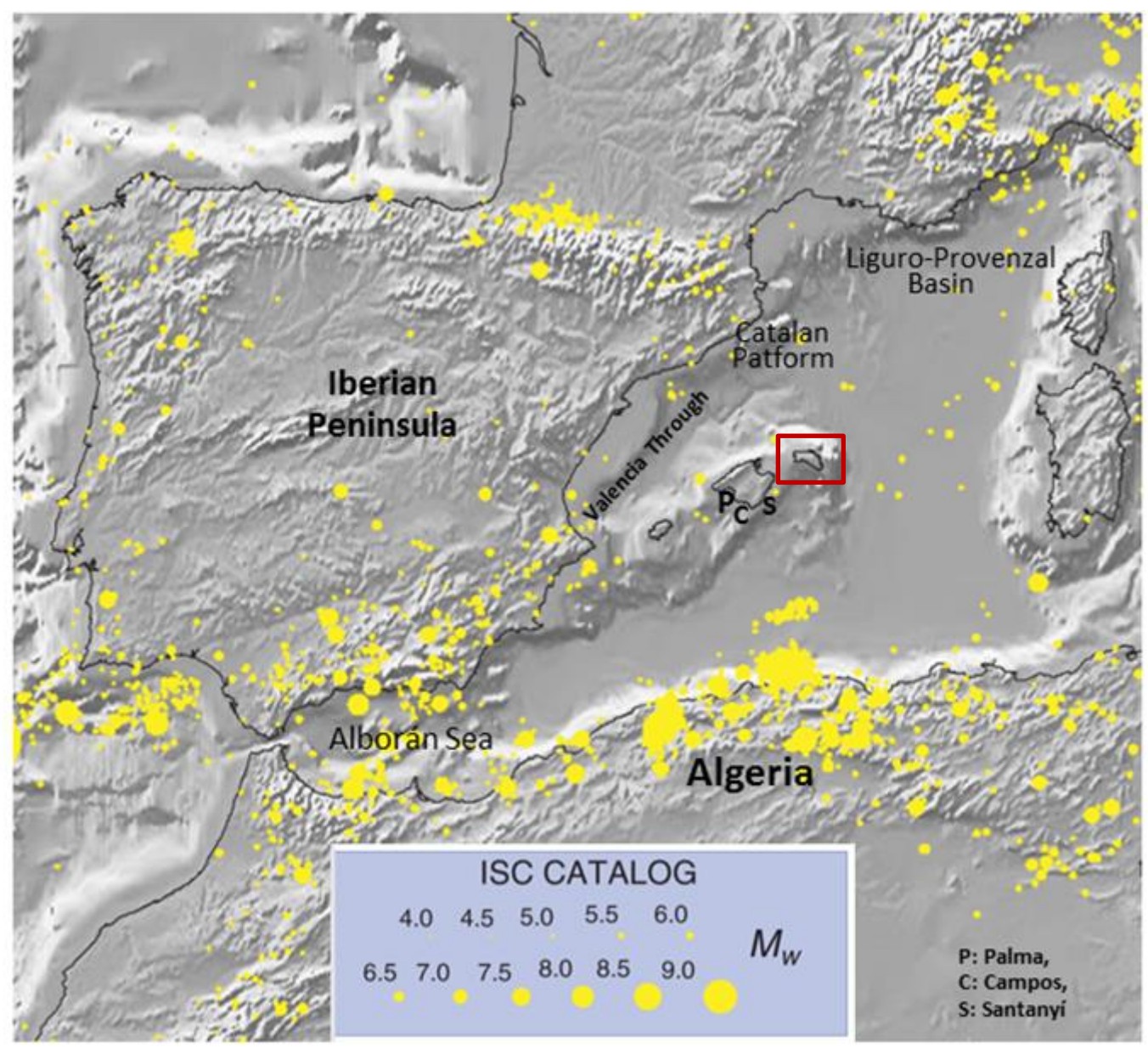

**Figure 2. Instrumental seismicity of the Western Mediterranean Region (from ISC Catalog) for depth interval 0-50 km. Modified from Vanucci et al., 2004. P refers to Palma, C refers to Campos and S refers to Santanyi, three sites mentioned in the text.**

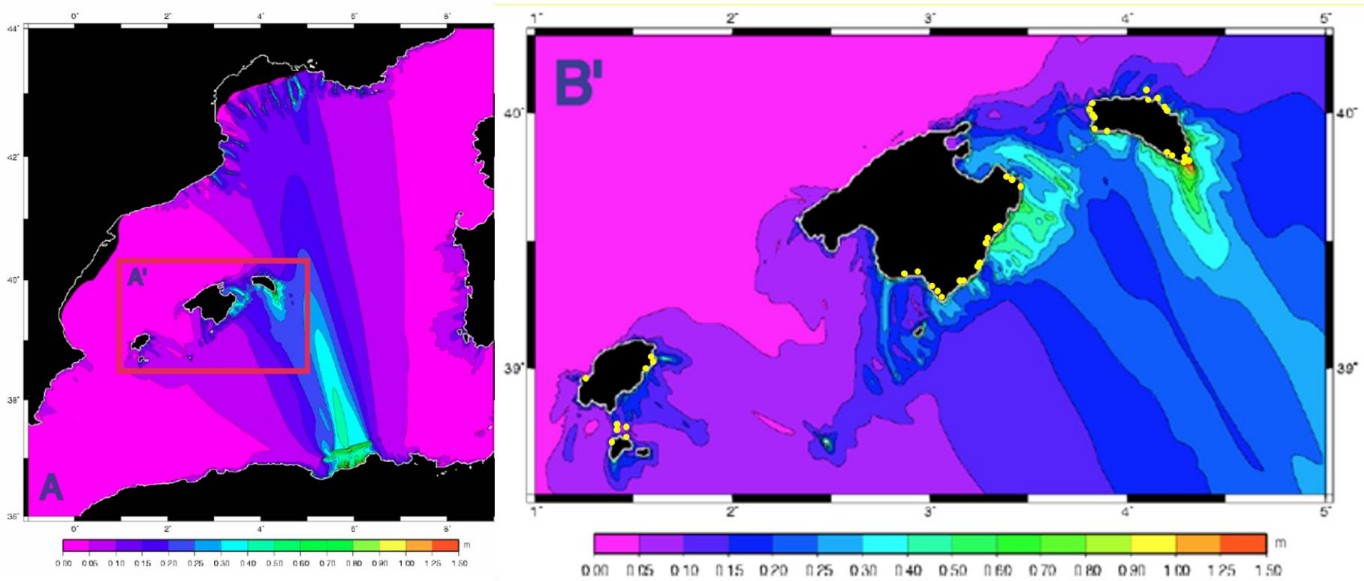

**Figure 3: Tsunami simulation, generated from a northern Algeria earthquake, impacting the Balearic Islands. Accumulated maximum height 1.5 h after the break of the fault, 3 segments at a time, with a deviation of 80 °. Source: Roger and Hebert (2008). Yellow dots correspond to study sites were boulders have been found. Note the correspondence between the simulation results and the location of the boulders.**

**Table 1. Historical tsunamis phenomena impacting in the Balearic Islands, modified from Roig-Munar (2016). Information sources (IS): (1) Fontseré (1918) and (2) Martinez-Solares (2001) and Silva and Rodriguez (2014) (see fig. 2, for location).**

| Data | Affected area | Phenomenon | IS |
|------|---------------|------------|-----|
| 1660 | Majorca, Palma, Campos | Earthquake and tsunami | 1 |
| 1721 | Balearic Islands | Earthquake and sea water withdrawal | 1 |
| 1756 | Majorca, Santanyí | Tsunami and big waves | 1 |
| 1756 | Balearic Islands | Tsunami and flooded coasts | 2 |
| 1790 | Alboran Sea | Tsunami | 2 |
| 1804 | Alboran Sea | Tsunami | 2 |
| 1856 | Minorca, Maó | Tsunami and seismic wave | 1 |
| 1856 | Algeria | Tsunami | 2 |
| 1885 | Algeria | Sea level changes | 2 |
| 1891 | Algeria | Tsunami | 2 |
| 1918 | Minorca, Maó | Seismic wave | 1 |
| 2003 | Algeria | Earthquake (7.0) and tsunami | 2 |

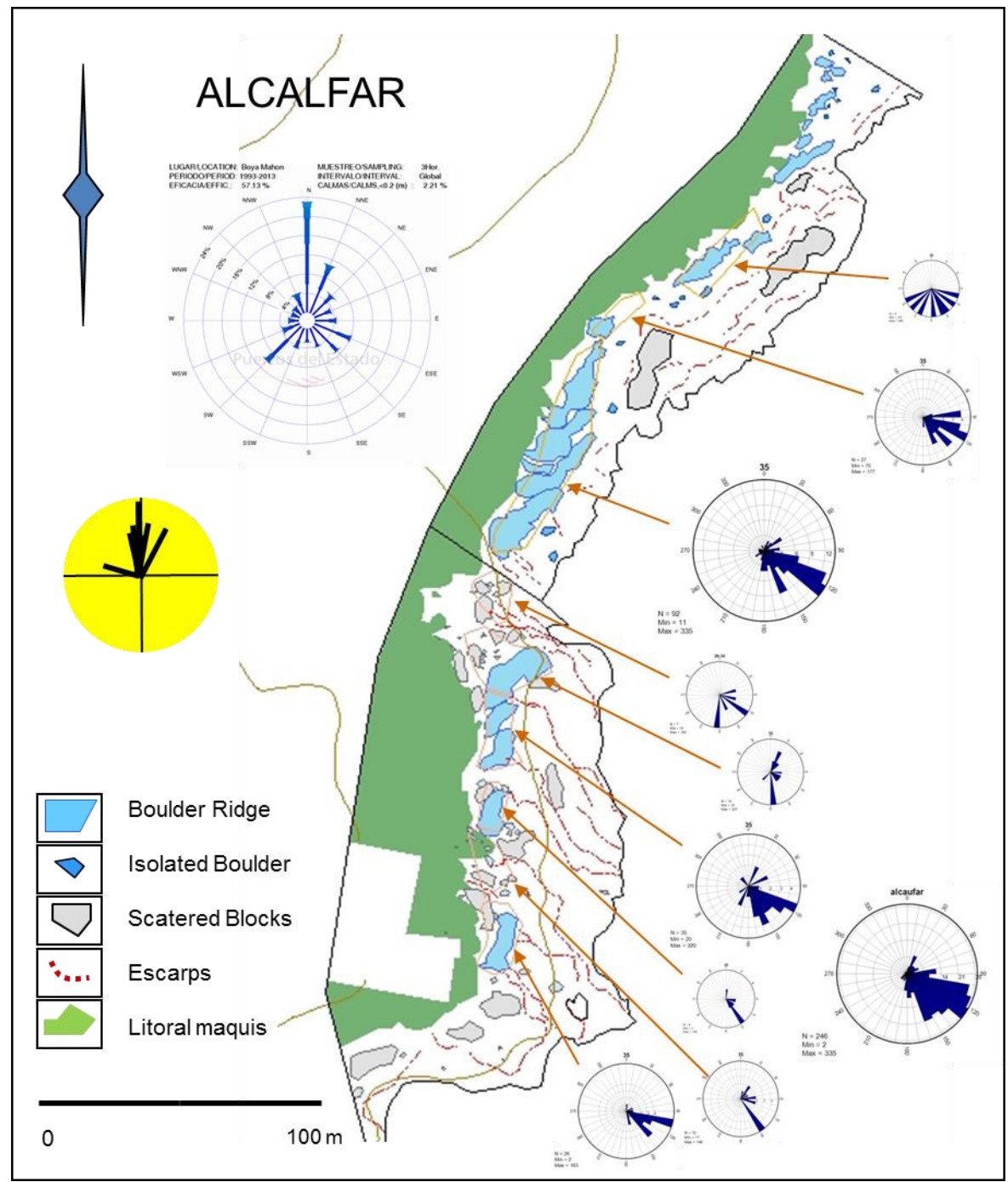

**Figure 4: Geomorphology Map of Alcalfar area (SE of Minorca) White circles show boulder orientation for each site. Main circle shows mean wave directions recorded at Maó Buoy. Yellow circle shows mean extreme wave directions**

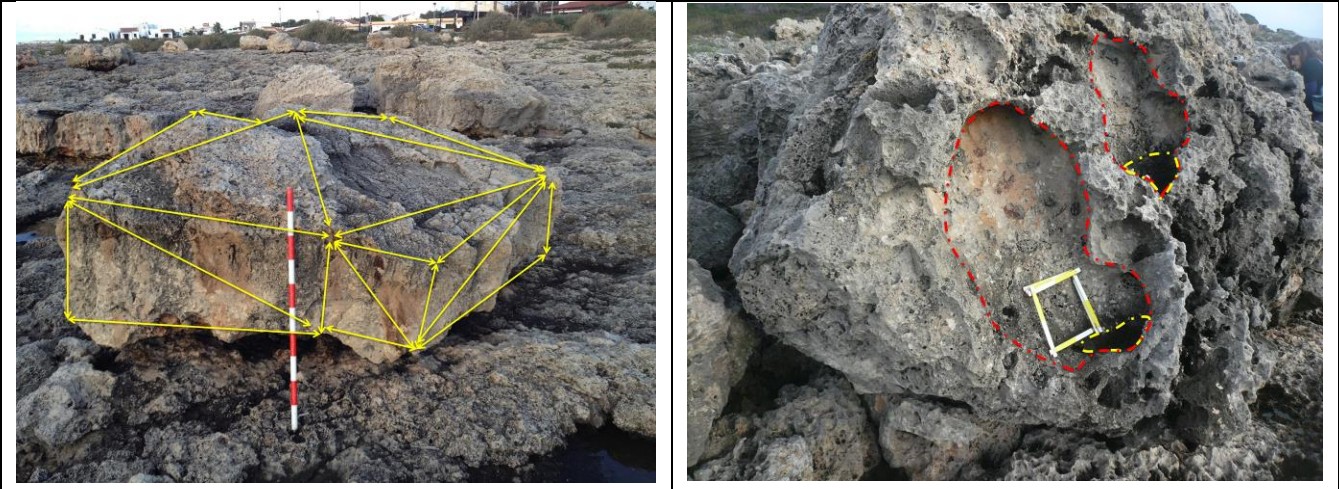

**Figure 5: a) Example of triangulation of a boulder to obtain the actual volume (*sa Caleta, Minorca*). b) Unconformable post-depositional morphologies (yellow) over pre-existing solution pans (red) (*son Ganxo, Minorca*).**

**Table 2: Equations used in the analysis of Minorca boulders**

| | | Ht | Hs |
|---|---|---|---|
| **Nott (2003)** | **submerged** | $Ht = [0{,}25(\rho_s - \rho_w / \rho_w ) 2a] / [(C_d (ac/b^2)+ C_l]$ | $Hs = [(\rho_s - \rho_w / \rho_w) 2a] / [(C_d (ac/b^2)+ C_l]$ |
| | **subaerial** | $Ht = [0{,}25 (\rho_s - \rho_w / \rho_w) [2a - C_m (a/b) (ü/g)] / [C_d (ac/b^2)+ C_l]$ | $Hs = [(\rho_s - \rho_w / \rho_w) [2a - 4C_m (a/b) (ü/g)] ] / [C_d (ac/b^2)+ C_l]$ |
| | **joint bounded boulder** | $Ht = [0{,}25 (\rho_s - \rho_w / \rho_w ) a] / C_l$ | $Hs = [(\rho_s - \rho_w / \rho_w ) a] / C_l$ |
| **Pignatelli (2009)** | **joint bounded boulder** | $Ht = [0{,}5 \cdot c \cdot (\rho_s - \rho_w / \rho_w )] / C_l$ | |
| **Engel and May (2012)** | **subaerial** | $Ht = 0{,}5 \cdot \mu \cdot V \cdot \rho_b / C_D \cdot (a \cdot c \cdot q) \cdot \rho_w$ | $Hs = 2 \cdot \mu \cdot V \cdot \rho_b / C_D \cdot (a \cdot c \cdot q) \cdot \rho_w$ |
| | **joint bounded boulder** | $Ht = (\rho_b - \rho_w) \cdot V \cdot (\cos\theta + \mu \cdot \sin \theta) / 2 \cdot \rho_w \cdot C_L \cdot a \cdot b \cdot q$ | $Hs = (\rho_b - \rho_w) \cdot V \cdot (\cos \theta + \mu \cdot \sin \theta) / 0{,}5 \cdot \rho_w \cdot C_L \cdot a \cdot b \cdot q$ |
| | Ht | tsunami height | a | large axis of the boulder | $C_d$ | coefficient of drag |
| | Hs | storm wave height | b | medium axis of the boulder | $C_l$ | coefficient of lift |
| | $\rho_s$ | boulder density | c | short axis of the boulder | $C_m$ | coefficient of mass |
| | $\rho_w$ | sea water density | g | force of gravity | ü | speed of water flow |
| | V | Volume abc of the boulder | q | boulder area coefficient | θ | cliff top steepness |
| | μ | coefficient of friction | | | | |

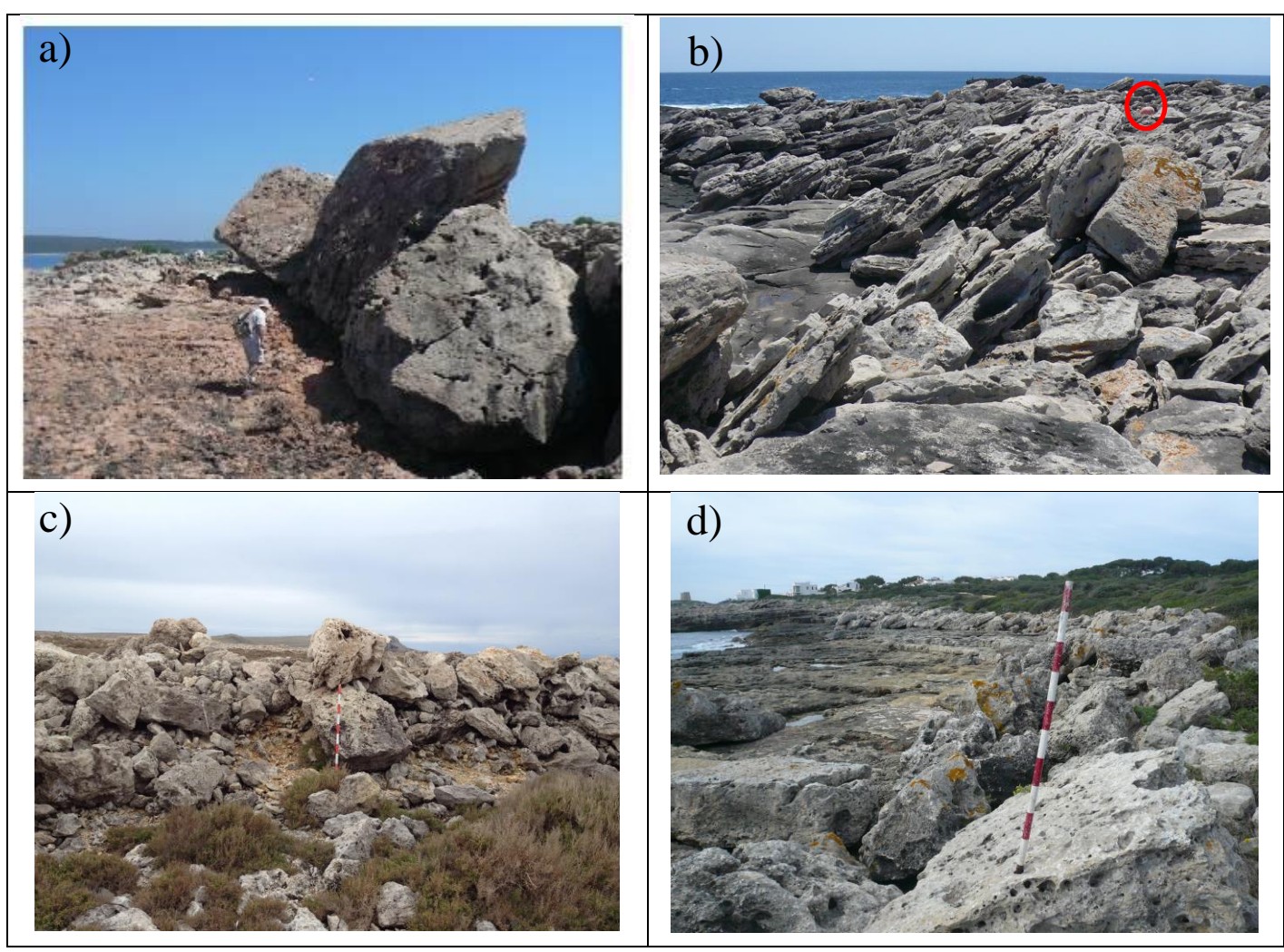

**Figure 6: a) Examples of mega-boulders displaced from the edge of the cliff at Illa de l'Aire, SE of Minorca, 15 m asl., b) Set of imbricate boulders at Sant Esteve, SE of Minorca, buoy in circle is 60 cm long c) Boulder ridge at Punta Nati, W of Minorca, 21 m asl. d) Ridge of imbricate boulders at Alcalfar, E of Minorca, 4.5 m asl. See fig 6 and 8 for location.**

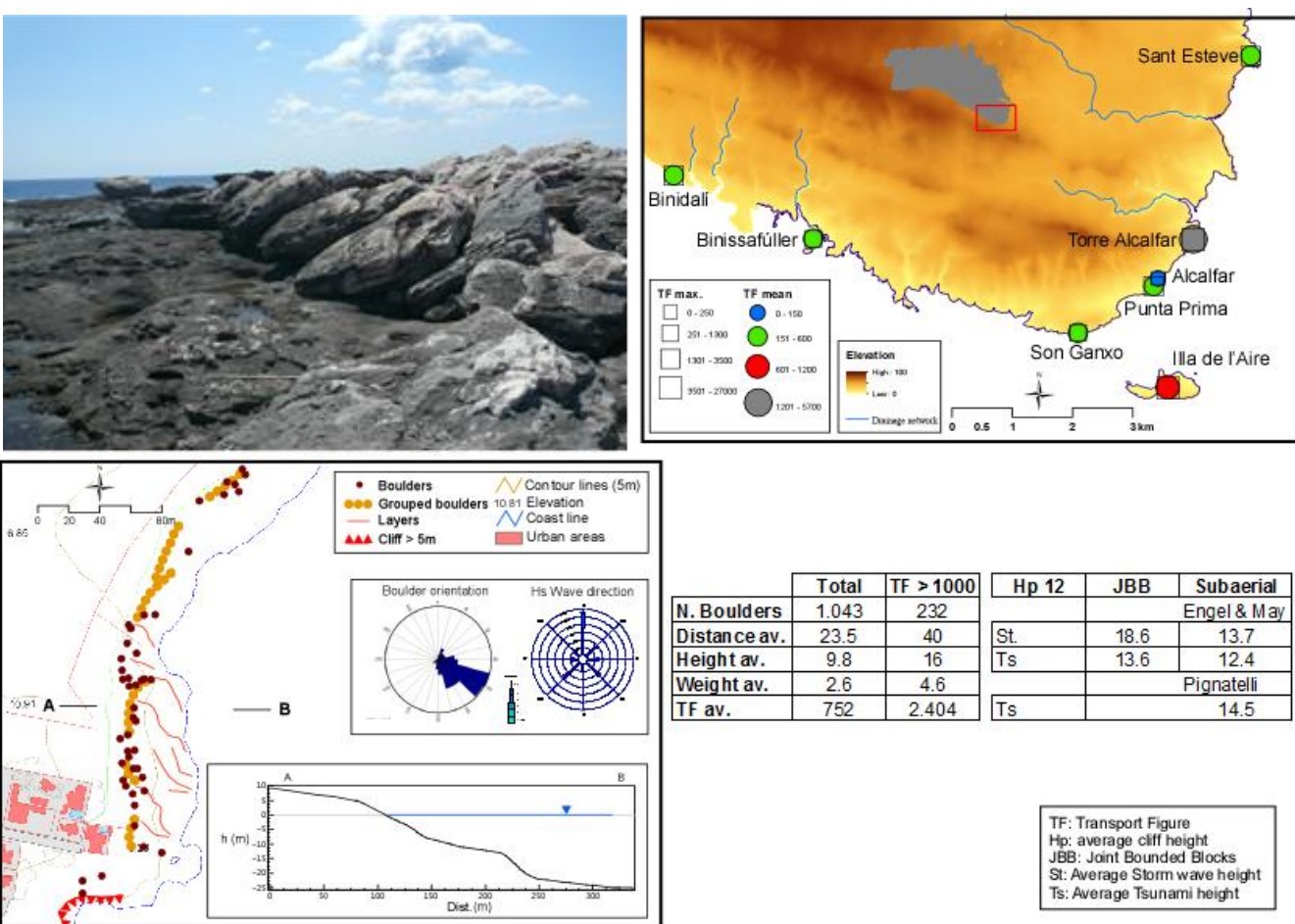

**Figure 7.- Location and main characteristics of SE Minorca boulders. Picture corresponds to an imbricate ridge of boulders in *Sant Esteve*. Geomorphological sketch shows boulders distribution at *Alcalfar*.**

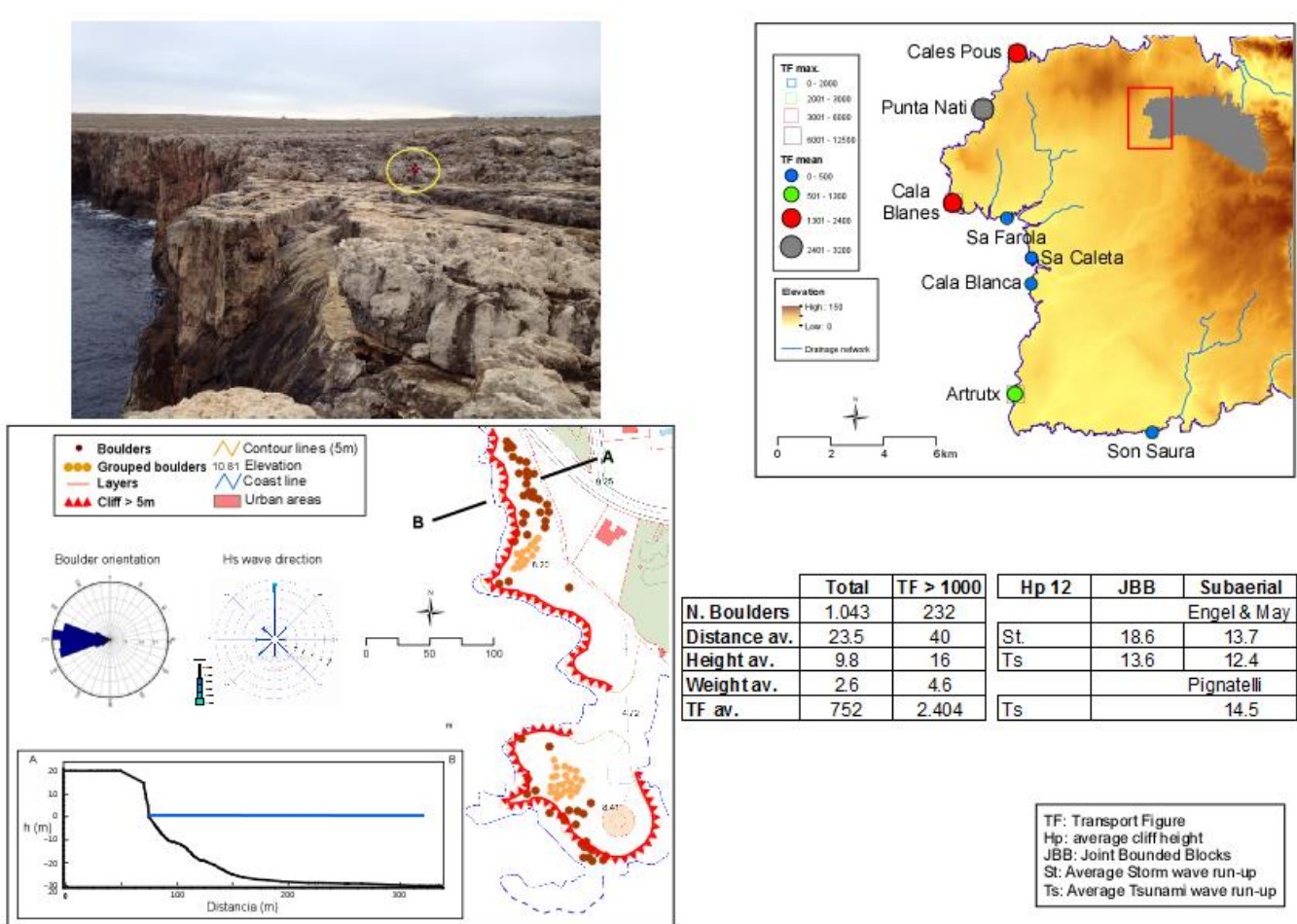

**Figure 8. Locations and main characteristics of W Minorca boulders. Picture corresponds to isolated boulders from *Punta Nati* (31 m above sea level). Geomorphological sketch shows boulders distribution at *Sa Caleta*..**

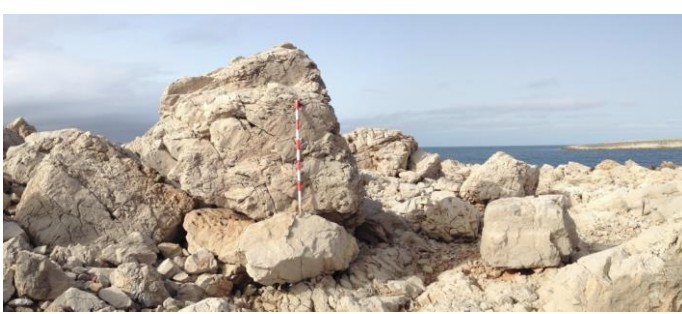

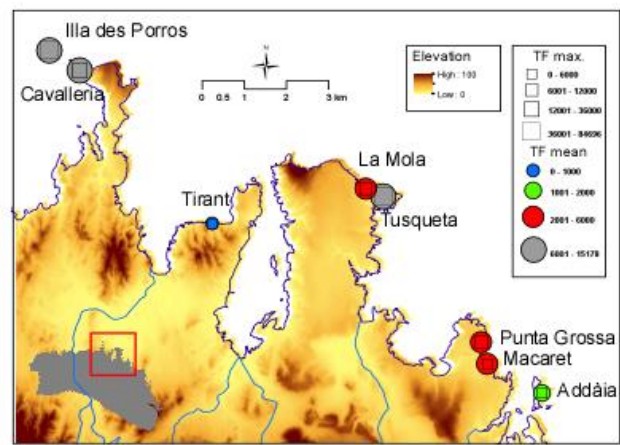

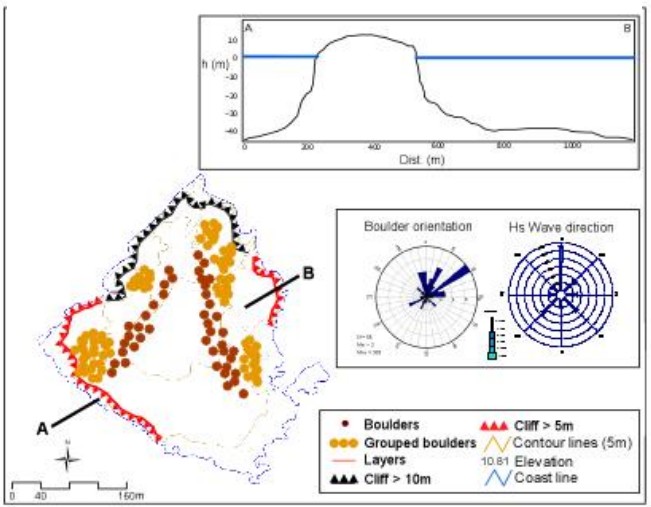

| | Total | TF > 1000 | Hp 7.8 | JBB | Subaerial |
|---|---|---|---|---|---|
| N. Boulders | 338 | 214 | | | Engel & May |
| Distance av. | 50.2 | 66.2 | St. | 21.6 | 9.8 |
| Height av. | 9.4 | 11.7 | Ts | 11.3 | 8.3 |
| Weight av. | 8.4 | 12.1 | | | Pignatelli |
| TF av. | 5.479 | 8.501 | Ts | | 13.2 |

TF: Transport Figure
Hp: average cliff height
JBB: Joint Bounded Blocks
St: Average Storm wave run-up
Ts: Average Tsunami wave run-up

**Figure 9. Location and main characteristics of N Minorca boulders. Picture corresponds to Caballeria boulders. Geomorphological sketch shows boulders distribution at Illot d'Addaia.**

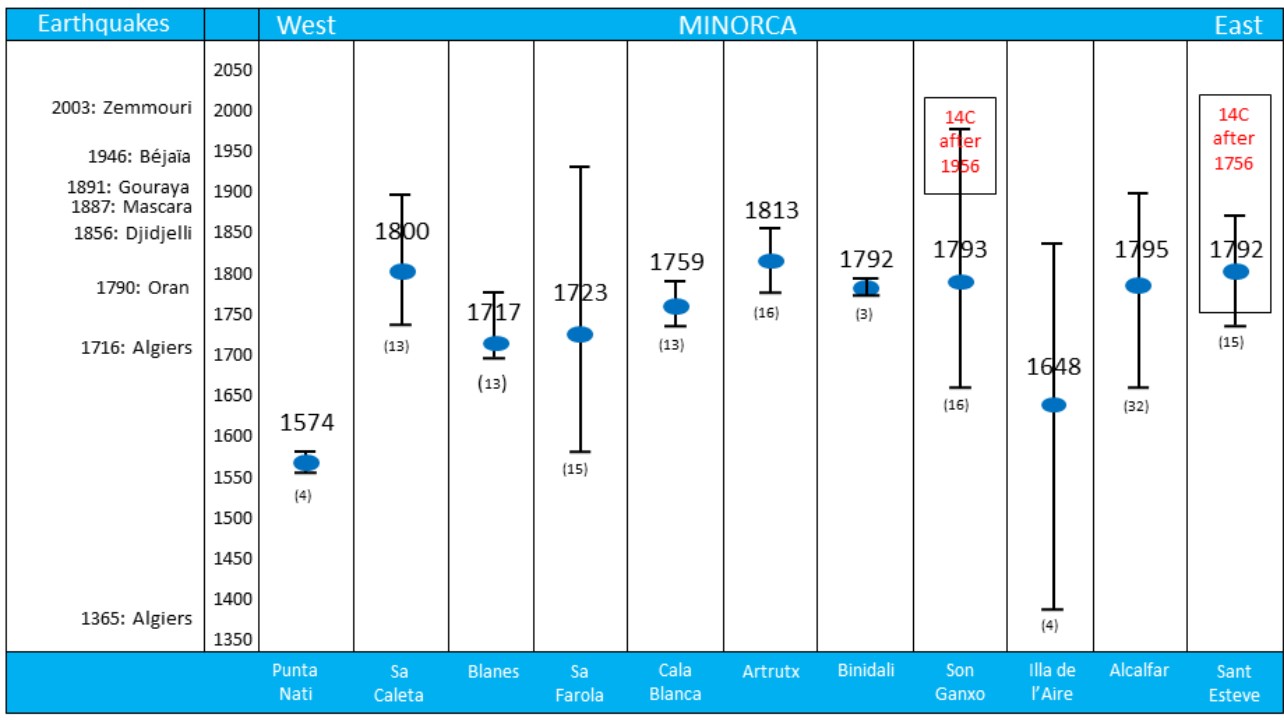

**Figure 10: Chronology of the post-depositional dissolution pans found on the surface of South Minorca boulders: The ages, in years AD, correspond to the post depositional dissolution pans measured on the boulders of the sampled localities. The blue dots indicate the average age of each locality. The bar indicates the range of dispersion of calculated ages, and the numbers in parentheses show the number of measured pans at each area. The left column displays the earthquakes with intensity >X occurred in North Algerian Coast, since 1365. Rectangles indicate the age obtained through 14C.**

**Table 3: Comparison for JBB with TF>1000 of calculated Storm Run-up, Tsunami Run-up, Transport Figure and Maximum wave height. (Hs) Average Storm wave height from Engel and May (2012), (Ht) Average Tsunami wave height from Engel and May (2012), (Alt) Average altitude of measured boulders, (Rs) Run-up needed for storm waves (Hs+Alt), (Rt) Run-up needed for tsunami waves (Hs+Alt), (TF) Transport Figure average and (Hm) Maximum wave height recorded by Cañelles (2007)**

| Sector | Hs | Ht | Alt | Rs | Rt | TF | Hm |
|--------|------|------|-------|-------|-------|-------|------|
| North | 13,79 | 3,45 | 7,81 | 21,60 | 11,26 | 8.501 | 11,0 |
| West | 6,61 | 1,66 | 11,97 | 18,58 | 13,63 | 2.404 | 8,0 |
| Southeast | 7,58 | 1,90 | 6,80 | 14,38 | 8,70 | 2.466 | 7,5 |