# Peer review of "Tsunamis boulders on the rocky shores of Minorca (Balearic Islands)"

_Natural Hazards and Earth System Sciences, 2017_

## Referee Comment (RC1) · Anonymous Referee #1 · 8 Oct 2017

The paper by Roig-Munar et al. is interesting in general. However, the manuscript has some major issues related to the presentation of the data and other more formal aspects. On the one hand the introduction is over-referenced for case studies in the Mediterranean. On the other hand, important references are omitted. Statements like "Sedimentary records of tsunamis generated off the North African coast have been identified along the rocky coastline of Minorca,. . ."(l.24-25) come without reference. This sentence is rather a conclusion of the paper already. Page 2 line 1 starts rather abruptly with information on seismicity of Algeria – leaving the reader alone why this would be important. References are missing here as well and also in line 5, line 11. The introduction concludes with the statement that Roig Munar (2016) [how do you spell your name? with or without dash?] already identified the boulders as tsunamigenic

and dated most of them. So what is the aim of this paper here? Later in the text you refer to Roig-Munar et al. (2017). How does the present study differs from this one. I realise that Roig-Munar (2016) is a (unpublished?) PhD-thesis but you treat it as published scientific results. You should clearly outline the aims of this particular study, which could well be the same aims as in the thesis. The method section is in part a discussion on volume estimation of boulders – and not a description of the methods that you applied in your study site. Which directly leads to the next problematic formal aspect of the manuscript: you have no paragraph in the study site. The reader does not get any information on the geology, tidal range (negligible?), wave regime, climate, tectonic setting. A description of the study site is mandatory. The result chapter starts with the statement that 24 areas were analysed. Which areas? The reader has no idea what you talk about. The height information on cliffs and boulders have no reference water level. What do talk about? Mean high tide? Mean sea-level? The SI-unit for a metric ton is "t" and not "T". "Since the boulders do not record a single tsunami run-up, these figures can be estimated for the latest and most intense tsunami run-up" (p.4, l. 9-10): this is very confusing, why do they not record a single tsunami run-up? Line 6: storm wave. 7.5m (reference?) contradicts line 12 (8.5m). Line 13: the boulders have been interpreted as. . . by whom? Or do you mean: we interpret these boulders as. . . .? The same statement occurs in the following paragraphs. Line 27: regional wave height 8m – reference is missing here. Chapter 3.5: how can you 14C date something to 1964 AD? We need a table of your dating results. The last paragraph in this section is not results but discussion. Chapter 4: I do not see a strong relationship with seismic activity. You have not shown this in the manuscript. In line 27 you write the following statement: Glacial deposits only register the largest tsunami". At the moment the conclusions are not backed by the data. We need information in the dating, historical seismicity, historical records of tsunami, etc. Good luck

---

## Referee Comment (RC2) · Anonymous Referee #2 · 18 Oct 2017

General comments

The Authors show several data on boulders deposits on Minorca Island concluding that they were emplaced by tsunamis. The paper could be interesting but methods and results are not well presented, and there are some observations contradicting the conclusion. Therefore, I suggest improving the manuscript showing missing information and reformulating the conclusion.

My first remark regards data presentation. The Authors do not describe the study sites and their surroundings: a map with western Mediterranean area showing seismogenic sources and earthquake distribution would help to understand the location of tsunamigenic areas around Balearic Islands. Moreover, why do you exclude Iberian earthquakes from tsunamigenic sources? No information is shown on tidal range, wave

regime, geology and tectonic setting (did your sites undergo to uplift or subsidence?).

My second remark concerns the data on the maximum wave heights related to historical tsunamis that hit the Minorca Island. Authors show in Table 1 tsunamis observed in the Balearic Islands and their surroundings in the last four centuries, while in Figure 2 tsunami from northern Algeria affecting Balearic Islands modelled by Roger and Hebert (2008). Both show maximum wave heights of 2 m and the May 21, 2003 tsunami was 3 m high; it had the highest tsunami waves recorded in recent years in the Balearic Islands. With reference to the studied boulders, Authors affirm that "Our findings along the higher cliffs of the W coastline, requires tsunamis run-ups 13 m high and / or storm run-ups of 18.6 m". Therefore, neither tsunamis nor storms can have emplaced the boulders you observed in the present coast profile, because in your historical data no tsunami caused waves 13 m high. Probably boulders were deposited when the shoreline was lower than today is. On the other hand, it is possible that storms had higher waves than you observed or that the deposition of boulders by tsunami/s occurred before your historical observation period.

The last remark regards your dating methods. How did you date with 14C boulders 1964 AD and 1856 AD? Usually the last three centuries are uncertain in 14C dating. These boulders seem to have been recently emplaced, because are among the "five of the analysed boulders showing marine fauna", therefore they are likely storm boulders. Please show your dating results with calibration and error. Also dating with post-depositional dissolution pans (Fig. 4b) seems not to be very careful. In fact, dissolution rate is not uniform and the range of dispersion of calculated ages makes the values overlapping.

Specific comments

Page 1. Lines 25: "Sedimentary records of tsunamis generated off the North African coast have been identified along the rocky coastline of Minorca, as inland boulders, in most cases, ripped off a cliff edge…." by whom? Page 2. Line 1: "Historical and

instrumental seismicity indicates that North of Algeria is exposed to relevant seismic hazard and risk". You are not dealing with Algeria but with Minorca Island. Describe the seimotectonic setting of your study area at local scale and in the general western Mediterranean background. Page 2. Line 9: "Alvarez et al. (2011) modelled tsunamis generated near the Balearic Islands". What does it mean "near the Balearic Islands"? Page 2. Line 15: "Tsunami generated by these sources arrive in 30 minutes to Formentera, . . . and 45 minutes to Minorca". This information can be useful for tsunami alert system but not for your study. What about run-up heights predicted on the Minorca coasts? This information can help you to understand if boulders were deposited by tsunamis. Page 3. Lines 26-27. "Transport age of 145 boulders from 12 locations was determined using a combination of these methods." You have just two radiocarbon dating. How did you use this combination? It is not clear what boulders were dated with radiocarbon and the age of the same boulders resulting from dating surface post-transport features. Page 4. Line 9. "Since the boulders do not record a single tsunami run-up", what do you mean? Page 4. Line15. "In many areas, their origin must be established by a confluence of different criteria", what do you mean? Page 4. Line 25. "The average boulder height is 16 m and 40 m from the edge of the cliff", do you mean distance from the edge? Page 4. Line 33. "The heights of the boulders of this coastal sector are out of the reach of storm waves, and should be interpreted as tsunami deposits". Why? You have not tsunami run-up so high and it is possible that storm data are incomplete. Page 5. Paragraph 3.4 Biggest boulders. No boulders described in this section could have been deposited by storms and tsunamis. How do you explain them? Maybe they were emplaced when littoral platform was lower or the sea level was higher. Page 6. Lines 13-15. "Among the historical records of huge wave phenomena that have affected the Balearic Islands, there are some episodes that can be attributed to tsunamis. In 1654, the chronicles written by Fontseré (1918), record a hurricane in the sea that crossed the island of Minorca, destroying the foundations of buildings and uprooting trees." I do not understand the 1654 is not a tsunami but a hurricane; therefore, it is likely that in the Balearic Islands some meteorological events

was bigger than the storms about you should discuss (?) in the paper. Use always Majorca or Mallorca Kelletal, Keletat = Kelletat check please Please show in a map all the locations mentioned in the text.

In addition, I agree with the reviewer1 comments and found them very helpful. If addressed appropriately, the paper could be improved significantly. Finally, a revision of written English would be welcomed.

---

## Author Comment (AC1) · 4 Dec 2017

After analyzing all the considerations of the reviewers we accept practically all of them, especially the most concrete ones that will undoubtedly improve the manuscript. In relation to the more general comments we want to express the following considerations:

We agree with the referees than it is essential to improve the geographic and geological framework in which the analyzed deposits are located. Very probably the current wording does not allow to correctly understand its situation to readers not familiar with the island of Menorca. Therefore, we propose to better describe the situation of the places where the studied boulders are located. We will include information about the geological structure, lithology and also the maritime climate (tidal-range, which is negligible in the Balearic Islands, and wave regime).

Another comment by the two reviewers refers to the seismicity of the western Mediterranean that allows understanding the relationship between tsunami-generating earthquakes and the position of the studied boulders. For this we propose to incorporate an illustrative map of the situation of the earthquakes and a list of historical tsunamis in the western Mediterranean in order to relate the tsunamis with the boulders emplacement.

[Figure]

Related to this point, we exclude Iberian earthquakes from tsunamigenic sources because the imbricate boulders we have found in Minorca, but also in Mallorca, Ibiza and Formentera (the rest of the Balearic Islands), are located at the south, south-east of these Islands. Only in Minorca we have found boulders in the N and W, but these places are coincident with places where refracted tsunami waves hit the coast at the numerical models simulations from earthquakes at N-Africa.

A third comment from both reviewers refers to the fact that both, the position of the boulders and the results of the hydrodynamic equations, require run-ups of the tsunami wave that multiply, between two and ten times, the models forecast heights of tsunami waves in the open sea. First of all, the run-up of tsunamis on vertical cliffs is several times higher than that occurring on low coastal areas (Bryant, 2014). Run-up is also enhanced due to several factors (Lekkas et al., 2011): 1) by the distance from the tsunami generation area (of only 300 km in our case), 2) by the narrowness of the continental shelf (as in Minorca), 3) by the fact than the tsunami propagation vector is almost perpendicular to the main shoreline direction, and 4) land morphology, characterized by vertical cliffs with entrances (calas). For these reasons we think than run-ups heights in Minorca are several times higher than tsunami wave heights.

Finally, the reviewers raise the problem of the dating of the quarry and transport of the boulders. Although only two blocks with embedded marine fauna, have been radiocarbon dated, such dates serve as a reference to the second dating method used. It is true that the results of the C14 dates have been incorrectly stated in the article. After reviewing the reports corresponding to these dates can only be stated in a case that it is a block moved younger than 1720 AD, and in the other case that was transported after 1964.

The second dating method used -complementary to the previous one- is an approximation based on an average dissolution rate of dissolution pans (karstic depressions kamenitza type). This requires to identify post-depositional dissolution pans, that is, that have been formed after the movement of the boulders. They can be formed on the same boulder once transported or on the denudation surface that results from the quarry of the boulder. A margin of error can be established based on the variability of the dissolution speed, which is not very high because the boulders are located away from the cliff edge, where dissolution speed is much higher. However in no case the resulting values (age values) can be compatible with marine levels different from the current one. Other similar boulders dated by Kelletat (2005) in the neighboring island of Mallorca, corresponds to ages between 565 AD and 1508 AD. Thus, we think the boulders we are dealing were transported in the last centuries, with a marine level equal to the present one.

As a general comment we want to say that these article has its origin in the PhD Thesis of Roig-Munar (2016), which is unpublished.

Referring to the specific comments, we accept all of them and we will try to better explain what we were trying to explain in the former paper. And of course, we will include a revision of the written English.

As an anecdote, during the period in which the article was under review, a severe storm, in November of 2017, has caused waves of up to 11 meters in the north of the island of Minorca. We have made a field campaign days after the storm and all the blocks we had marked in advance (even in the ones which are only 1 m above sea level) have not moved, neither new blocks have been created. In the other hand, the tsunami that took place on May 21, 2003, generating 3m high waves, affecting the Balearic Islands, caused flooding in several calas (small beaches) in the east of Minorca (as stated by local newspapers), finding fishes hundreds of meters inland. Unfortunately, we did not study the blocks at that time.

References cited:

Bryan, E. 2014. Tsunami. The Underrated Hazard. Springer. 222 pgs.

Kelletat, D. 2005. Neue Beobachtungen zu Palao-Tsunami im Mittelmeergebeit, Mallorca und Bucht von Alanya, turkische Suudkuuste. Schriften des Arbeitskreises Landes- und Volkskunde Koblenz (ALV) 4, 1-14.

Lekkas, E., Andreadakis, E., Kostaki, I., Kapourani, E. 2011. Critical factors for Run-up and Impact of the Tohoku Earthquake Tsunami. International Journal of Geosciences, 2, 310-317.

---

## Author Response (AR1)

**"Tsunamis boulders on the rocky shores of Minorca (Balearic Islands)" by Francesc X. Roig-Munar et al.**

**Anonymous Referee #1**

The paper by Roig-Munar et al. is interesting in general. However, the manuscript has some major issues related to the presentation of the data and other more formal aspects. On the one hand the introduction is over-referenced for case studies in the Mediterranean. On the other hand, important references are omitted.
*We have removed the references regarding the eastern Mediterranean and we have included new, more relevant and general references (distributed throughout the text).*

Statements like "Sedimentary records of tsunamis generated off the North African coast have been identified along the rocky coastline of Minorca,..(l.24-25) come without reference. This sentence is rather a conclusion of the paper already.
*We agree. We have changed this sentence.*

Page 2 line 1 starts rather abruptly with information on seismicity of Algeria – leaving the reader alone why this would be important. References are missing here as well and also in line 5, line 11.
*We have changed this point, reorganizing almost the entire introduction and explaining why seismicity in Algeria is important for the understanding of the presence of this coastal boulders in Minorca.*

The introduction concludes with the statement that Roig Munar (2016) [how do you spell your name? with or without dash?] already identified the boulders as tsunamigenic and dated most of them. So what is the aim of this paper here? Later in the text you refer to Roig-Munar et al. (2017). How does the present study differs from this one. I realize that Roig-Munar (2016) is a (unpublished?) PhD-thesis but you treat it as published scientific results. You should clearly outline the aims of this particular study, which could well be the same aims as in the thesis.
*The referee is wright. Roig-Munar (2016) is the unpublished PhD Thesis of the first author of this article. We have removed other little and local publications of the author*

*because are not relevant. We also included some new outcrops to complete the Minorca description of these deposits.*

The method section is in part a discussion on volume estimation of boulders – and not a description of the methods that you applied in your study site. Which directly leads to the next problematic formal aspect of the manuscript: you have no paragraph in the study site. The reader does not get any information on the geology, tidal range (negligible?), wave regime, climate, tectonic setting. A description of the study site is mandatory.

*We created a new section called "Study site" explaining the geological framework and the maritime climate of Minorca.*

The result chapter starts with the statement that 24 areas were analyzed. Which areas? The reader has no idea what you talk about.

*We described the results of these areas grouped in tree sectors and display their location in the figures.*

The height information on cliffs and boulders have no reference water level. What do talk about? Mean high tide? Mean sea-level?

*Height information refers to mean sea level. Tide amplitude is negligible at this effect.*

The SI-unit for a metric ton is "t" and not "T".
*Corrected.*

"Since the boulders do not record a single tsunami run-up, these figures can be estimated for the latest and most intense tsunami run-up" (p.4, l. 9-10): this is very confusing, why do they not record a single tsunami run-up?
*We agree. Is confusing. We removed that sentence.*

Line 6: storm wave. 7.5m (reference?) contradicts line 12 (8.5m).
*Corrected.*

Line 13: the boulders have been interpreted as ... by whom? Or do you mean: we interpret these boulders as...? The same statement occurs in the following paragraphs.
*We interpret these boulders is correct.*

Line 27: regional wave height 8m – reference is missing here.
*We introduced the reference.*

Chapter 3.5: how can you 14C date something to 1964 AD? We need a table of your dating results. The last paragraph in this section is not results but discussion. We agree.
*The last paragraph is in the discussion now. Referee was right, according to 14C date the age is younger than 1964 AD.*

Chapter 4: I do not see a strong relationship with seismic activity. You have not shown this in the manuscript.
*Now seismic activity and boulder setting relationship is shown in Figures 9 and 10.*

In line 27 you write the following statement: Glacial deposits only register the largest tsunami".
*We removed that sentence.*

At the moment the conclusions are not backed by the data. We need information in the dating, historical seismicity, historical records of tsunami, etc. Good luck
*Now in the article dating, historical seismicity and historical records are more clearly explained.*

**Anonymous Referee #2**

General comments

The Authors show several data on boulders deposits on Minorca Island concluding that they were emplaced by tsunamis. The paper could be interesting but methods and results are not well presented, and there are some observations contradicting the conclusion. Therefore, I suggest improving the manuscript showing missing information and reformulating the conclusion.

My first remark regards data presentation. The Authors do not describe the study sites and their surroundings: a map with western Mediterranean area showing seismogenic sources and earthquake distribution would help to understand the location of tsunamigenic areas around Balearic Islands. Moreover, why do you exclude Iberian earthquakes from tsunamigenic sources? No information is shown on tidal range, wave regime, geology and tectonic setting (did your sites undergo to uplift or subsidence?).
*We present in the new version the seismogenic sources and earthquake distribution in the areas around the Balearic Islands. Iberian earthquakes are of too low magnitude to generate tsunamis reaching the Balearic coasts. We also exclude Iberian earthquakes from tsunamigenic sources because the imbricate boulders we have found in Minorca, but also in Mallorca, Ibiza and Formentera (the rest of the Balearic Islands), are mostly located at the south, south-east of these Islands. Only in Minorca we have found boulders in the N and W, but these places might be beaten by refracted tsunami waves according to the numerical models simulations from earthquakes at N-Africa. We explain that in the text.*
*Maritime climate have been also included in the text.*

My second remark concerns the data on the maximum wave heights related to historical tsunamis that hit the Minorca Island. Authors show in Table 1 tsunamis observed in the Balearic Islands and their surroundings in the last four centuries, while in Figure 2 tsunami from northern Algeria affecting Balearic Islands modelled by Roger and Hebert (2008). Both show maximum wave heights of 2 m and the May 21, 2003 tsunami was 3 m high; it had the highest tsunami waves recorded in recent years in the Balearic Islands. With reference to the studied boulders, Authors affirm that "Our findings along the higher cliffs of the W coastline, requires tsunamis runups 13 m high and / or storm run-ups of 18.6 m". Therefore, neither tsunamis nor storms can have emplaced the boulders you observed in the present coast profile, because in your historical data no tsunami caused waves 13 m high. Probably boulders were deposited when the shoreline was lower than today is. On the other hand, it is possible that storms had higher waves than you observed or that the deposition of boulders by tsunami/s occurred before your historical observation period.

*According to our data, the position of the boulders and the results of the hydrodynamic equations require tsunami wave run-ups that multiply, between two and ten times the forecast heights of tsunami waves in the open sea. First of all, the run-up of tsunamis on vertical cliffs is several times higher than that occurring on low coastal areas (Bryant, 2014). Run-up is also enhanced due to several factors (Lekkas et al., 2011): 1) by the distance from the tsunami generation area (only 300 km in our case), 2) by the narrowness of the continental shelf (as in Minorca), 3) by the fact than the tsunami propagation vector is almost perpendicular to the main shoreline direction, and 4) land morphology, characterized by vertical cliffs with entrances (calas). For these reasons we think than run-ups heights in Minorca are several times higher than tsunami wave heights.*

The last remark regards your dating methods. How did you date with 14C boulders1964 AD and 1856 AD? Usually the last three centuries are uncertain in 14C dating. These boulders seem to have been recently emplaced, because are among the "five of the analyzed boulders showing marine fauna", therefore they are likely storm boulders. Please show your dating results with calibration and error. Also dating with post-depositional dissolution pans (Fig. 4b) seems not to be very careful. In fact, dissolution rate is not uniform and the range of dispersion of calculated ages makes the values overlapping.

*After reviewing the reports corresponding to these dates can only be stated in a case a block moved after 1720 AD, and in the other case it was transported after 1964.*

*The second dating method used is based on the average dissolution rate of dissolution pans. This requires to identify post-depositional dissolution pans, that is, that have been formed after the movement of the boulders. They can be formed on the same boulder once transported or on the denudation surface that results from the quarry of the boulder. A margin of error can be established based on the variability of the dissolution rate, which is not very high because the boulders are located quite away from the cliff edge, where dissolution rate and their variability is much higher. However in no case the resulting age values can be compatible with marine levels*

*different from current sea level. Other similar boulders dated by Kelletat (2005) in the neighboring island of Mallorca, corresponds to ages between 565 AD and 1508 AD. Thus, we think the boulders we are dealing were transported in the last centuries, with a marine level equal to the present one.*

Specific comments
Page 1. Lines 25: "Sedimentary records of tsunamis generated off the North African coast have been identified along the rocky coastline of Minorca, as inland boulders, in most cases, ripped off a cliff edge...." by whom?
*We have changed that sentence.*
Page 2. Line 1: "Historical and instrumental seismicity indicates that North of Algeria is exposed to relevant seismic hazard and risk". You are not dealing with Algeria but with Minorca Island. Describe the seismotectonic setting of your study area at local scale and in the general western Mediterranean background.
*Now the general seismotectonic setting of the western Mediterranean and its relationship with the setting of Minorca boulders is more clearly described in the article.*
Page 2. Line 9: "Alvarez et al. (2011) modelled tsunamis generated near the Balearic Islands". What does it mean "near the Balearic Islands"?
*We removed that sentence.*
Page 2. Line 15: "Tsunami generated by these sources arrive in 30 minutes to Formentera and 45 minutes to Minorca". This information can be useful for tsunami alert system but not for your study. What about run-up heights predicted on the Minorca coasts? This information can help you to understand if boulders were deposited by tsunamis.
*We agree with the referee. We removed that sentence.*

Page 3. Lines 26-27. "Transport age of 145 boulders from 12 locations was determined using a combination of these methods." You have just two radiocarbon dating. How did you use this combination? It is not clear what boulders were dated with radiocarbon and the age of the same boulders resulting from dating surface post-transport features.
*The two methods are actually independents, but both describe boulders ages just some decades or centuries from present.*

Page 4. Line 9. "Since the boulders do not record a single tsunami run-up", what do you mean?
*This sentence have been removed.*

Page 4. Line15. "In many areas, their origin must be established by a confluence of different criteria", what do you mean?
*First we have to discard the blocks coming from gravitational crashes or those clearly wrenched by the waves, then we have to describe their position regarding the morphology and characteristics of the cliff.*

Page 4. Line 25. "The average boulder height is 16 m and 40 m from the edge of the cliff", do you mean distance from the edge?
*Yes, we describe the altitude of the boulder above mean sea level and their distance from cliff edge.*

Page 4. Line 33. "The heights of the boulders of this coastal sector are out of the reach of storm waves, and should be interpreted as tsunami deposits". Why? You have not tsunami run-up so high and it is possible that storm data are incomplete.
*Apart from the increase in the run-up due to the impact of the tsunami on the cliff, it is necessary to consider the location of the blocks and the direction of the imbrications, coinciding with the areas of lower swell of Minorca*

Page 5. Paragraph 3.4 Biggest boulders. No boulders described in this section could have been deposited by storms and tsunamis. How do you explain them? Maybe they were emplaced when littoral platform was lower or the sea level was higher.
*The ages described make it impossible to consider the possibility of different sea levels than the current one. In addition, the run-up obtained for these blocks are compatible with the impact on cliffs of modeled tsunamis.*

Page 6. Lines 13-15. "Among the historical records of huge wave phenomena that have affected the Balearic Islands, there are some episodes that can be attributed to tsunamis. In 1654, the chronicles written by Fontseré (1918), record a hurricane in the sea that crossed the island of Minorca, destroying the foundations of buildings and uprooting trees." I do not understand the 1654 is not a tsunami but a hurricane; therefore, it is likely that in the Balearic Islands some meteorological events was bigger than the storms about you should discuss (?) in the paper.

*Already discussed. Even a new word -Medicane- have been created for this events, but waves from medicanes do not reach the altitudes and fluxes needed for quarry and transport analyzed boulders.*

5 Use always Majorca or Mallorca Kelletal, Keletat = Kelletat check please.
*Corrected.*

Please show in a map all the locations mentioned in the text.
*Corrected.*

In addition, I agree with the reviewer1 comments and found them very helpful. If addressed appropriately, the paper could be improved significantly.
*We hope that this purpose has been achieved*

15 Finally, a revision of written English would be welcomed.
*It is done too.*

[revised manuscript text omitted]

---

## Author Response (AR2)

**General comments**

We agree with most of the general comments. In fact, we have reorganized the chapters according to the suggestions of the Referee # 2. We have moved fragments of the text that fit better in the Study site chapter according to these indications. In particular we improve the description of the Maritime climate including the description of recent tsunamis affecting Minorca and the Balearic Islands. We also accept to move the figure of earthquake distribution (Fig. 10) to the first part of the paper (now is figure 2).

We consider very meaningful the distribution of boulder sites at the Balearic Islands (figures 1 and 3). Boulders sites in Mallorca are distributed along the eastern and southern coast and the same happens in Ibiza. Only in Minorca we found boulder sites at the north coast, despite most of the boulder settings are located in the south and west coasts of the island. In figure 3 we show the perfect correspondence between the expected locations where a northern Africa generated tsunami will hit (from numerical model simulation) and the sites where the authors have found boulders accumulations. Why we haven't found large boulders at the western and northern coast of Mallorca or at the northern coast of Ibiza, with similar geological features, if wind fetch is the largest in this direction?. The answer is because the boulders are tsunami related.

The second general remark of referee #2 and many of its specific comments are dealing with run-ups of storm waves vs tsunami wave run-ups. We agree with its considerations about storm waves and their run-ups as they shoal, but we don't agree about tsunami wave run-ups. In fact, the run-up of tsunami waves differs absolutely when tsunami hits on low shores that when it does on cliffs, where it is not possible any run-in until tsunami wave overcome the cliff edge. Storm wave run-ups can increase the wave height in (as a maximum) a factor between 2 and 3 times, meanwhile in the cliff run-ups the increase factor reach up to 10 (Lekkas et al., 2011) to 40 times the tsunami wave height (as described in Hawaii; SMS Tsunami Warning web page). Moreover when tsunami source is so close to Minorca southern shores.

The last remark regards the dating methods. We agree with the referee about the significance of our results: they just remarks that boulders where dislodged and transported in recent times. This result rules out the interpretation of the boulder ridges as old coastlines. But on the other side, they don't support any interpretation as storm wave transport, neither as tsunami wave movement. Again, our interpretation as tsunami transported boulders is based not in their estimated age, but in their setting. Most boulders settings are located facing to the South, which is towards the main tsunamitic sources of this part of the Mediterranean. Moreover, main storm waves reach Minorca from the North (the fetch for this northern direction is of 700 km), meanwhile the southern storms are weaker because of its reduced fetch (lower to 300 km).

On the contrary, tsunami sources are clearly located offshore the Algerian coasts (Fig.2) and although some submarine slides have been described offshore the Ebro delta and the Southern coast of France, recent (2003) and historical dates point out the southern provenance of the main events.

We also disagree with the referee about the interpretation of our results of dating with post-depositional pans (Fig. 10): the dispersion of the results is due to the variability of the range of dissolution and can not be attributed only to the possibility they were transported by storm wave run-ups.

Boulder ridges, (Fig. 6) and Transport Figures values are, in our opinion, clearly evidences of tsunami transport, however storm wave actions are present in some of the lower settings.

**Specific comments.**

We have accepted most of the specific comments, with the exception of the ones regarding with the run-up interpretations as we had already described.

Page 1, Lines 23: "Some are positioned well above the maximum stand of any recorded storm wave." *Suppressed.*

Page 1, Line 30: "In fact, in many areas of the Western Mediterranean, metric size boulders have been interpreted as remnants of the tsunamis occurred in the last centuries (Pignatelly = Pignatelli et al., 2009). Yes, but other authors showed that these are storms deposited (again you should demonstrate)". *Out of range of this paper*

Page 2, Lines 1-26: "Move all this part in the chapter where you describe the setting of your region and the maritime climate". *Accepted.*

Page 2, Line 20: "There are also historical records reporting a flooding event with a run-in up to 2 km inland on the east coast of Majorca (the largest of the Balearic Islands) in 1756 (Fontsere, 1918)". Reformulate this sentence: There are also historical tsunami records reporting a flooding up to 2 km inland on the east coast of Majorca (Yes but in Minorca?). *Accepted.*

Page 2, Lines 7-8: "The last seismic event recorded that affected Minorca Island was the Zemmouri (Algeria) earthquake that took place on May 21, 2003, with a magnitude of 6.9 Mw". Was it the earthquake that affected Minorca or rather Minorca was affected by the following tsunami?. *Accepted. Just the tsunami generated by the Zummari earthquake affected Minorca.*

Page 2, Line 25: "Thus, in the last 60 years the maximum extremal wave height detected is of 11 m at the 2001 medicane (Jansà, 2013)". See general comments, was it observed at buoy?". *Yes, all the storm wave data comes from deep water buoys.*

Page 4, Line 20: "in the last 50 years by a maximum wave height of 10 m (use always 60 years or 50 years)". *Accepted. Wave data correspond to 50 years according to Cañelles (2010).*

Page 5, Line 20: "blocs = blocks". *Accepted.*

Page 7, Lines 29-30: "storm run-ups of 14 m are needed to dislodge the boulders, while tsunamis run-ups of only 8 and 13 m would explain their position". The observed maximum tsunami wave is 3 m and it is 10 m less than the run-up that need to dislodge the boulders (13 m), whereas the 14 m storm run-up is only 3 m more than the 11 m observed!". *See the general comments for run-up discussion.*

Page 8, Lines 2-4: "tsunamis run-ups 13 m high and/or storm run-ups of 18.6 m. ….and require storm run-ups of more than 21 m that are not plausible, while the height of a tsunami run-up required to position the boulders is

only 9 meters." Are you sure? 13 m vs 18.6 m and 9 m vs 21 m, some calculation was wrong. *See the general comments for run-up discussion*.

Page 8, Lines 12-14: "For these reasons we think than run-ups heights on Minorca would have been several times higher than tsunami wave heights. On the contrary, as they shoals, wave heights increase its run-up heights in a much lesser way and thus, it is impossible to reach the run-up values obtained from the hydrodynamic equations". Very confusing sentence. Furthermore, you must not think but demonstrate, for example computing run-up at coast using values at deep water. *See the general comments for run-up discussion*.

Page 8, Lines 28-30: "Regarding the dating of the boulders, although only two blocks with embedded marine fauna have been radiocarbon dated, such dates serve as a reference to the second dating method used. Our C14 results show than in one case a block was moved after 1720 AD, Sure? Your dating was 1856 AD. Was not it?". *Accepted. We have included all the details from C14 data.*

Page 9, Lines 25-26: "have been dislodged and positioned by the action of tsunami waves, although some of these boulders have also been reworked by storm waves". I do not understand. Why can storms rework boulders but cannot deposit them?. *Accepted. Boulder reworking can be considered in some way a form of deposition.*

Specific comments of figures and table all accepted

[revised manuscript text omitted]

---

## Author Response (AR3)

**EDITOR'S COMMENTS**

**Editor Decision: Reconsider after major revisions (further review by editor and referees)** (06 Apr 2018) by Mauricio Gonzalez

Comments to the Author:

Dear authors,

After the second review iteration, the reviewer2 believe that this manuscript is not ready yet to be accepted for publication, next you have the comments:

I have seen that authors have addressed many of the requests, although they have glossed over many others by using this smart sentence "See the general comments for run-up discussion" I hope they go in deep on these based on my previous revision and comments.

*We now go in deep, point by point, to the comments of referee 2.*

Moreover, Authors should check carefully references and order of figures and separating discussion on tsunamis from the paragraph 2.2. Maritime climate in other section.

*We have separated tsunami discussion from maritime climate.*

In order to avoid more iterations, I ask the authors to take into account all the comments addressed by the reviewer2 (last review and the previous one). Based in this reply I will make a final decision to accept or reject the paper for publication.

*We now proceed to take into account all the comments addressed by reviewer 2, even the already answered in the previous review.*

**GENERAL COMMENTS (Referee 2)**

**1) Authors** start affirming that most boulders are deposited by tsunamis, but this is the aim of the paper that should be demonstrated. Usually when writing a paper, some rules that help readers to understand are listed as follow:

1) In the introduction, what is already known and then what we want to demonstrate (and not the conclusion we want).

2) General geological framework with information on seismic sources and tsunamigenic earthquakes and observed tsunamis

2.3) Study site

3) Method

4) Results

5) Discussion

6) Conclusion

Therefore, I suggest reordering the chapters.

5 *We have reorganized the chapters according to the suggestions of the Referee # 2. We have moved fragments of the text that fit better in the Study site chapter according to these indications. In particular we improve the description of the Maritime climate including the description of recent tsunamis affecting Minorca and the Balearic Islands.*

**2) Abstract** "The age of the boulders in most of the studied localities show a good correlation with historical 10 tsunamis. Age of the boulders, direction of imbrication and estimation of run-up necessary for their placement, indicate dislodging and transport by North African tsunami waves that hit the coastline of Minorca".

You have not shown a good correlation with historical tsunamis, age is either recent or post some date; furthermore, you did not estimated true possible run-up, but run-up that need to move boulders.

*We changed that sentence. Our data just remarks that boulders where dislodged and transported in recent times 15 and this result rules out the interpretation of the boulder ridges as old coastlines (this is also an important conclusion).*

*Historical data (Fontsere -1918-) reports a tsunami flooding event with a run-in of to 2 km inland in Santanyí (location on Fig. 2), on the east coast of Majorca, in 1756. Two kilometers inland in Cala Santanyí means a run-up of 45 m, calculated from topographic maps.*

**3) The map with western Mediterranean** area showing earthquake distribution would help to understand the location of tsunamigenic areas around Balearic Islands. This map is the last figure, but it should be the first.

*We moved the figure of earthquake distribution (Fig. 10) to the first part of the paper (now is figure 2).*

Moreover, why do you exclude Iberian earthquakes from tsunamigenic sources, if the 1756 tsunami was observed 25 at Balearic Islands? In the first version of the manuscript, this was clearly reported and you deleted it.

*We do not exclude Iberian earthquakes from tsunamigenic sources but the seismicity is very low at the Valencia Through (Figure 2) and the Alboran region is dominated by strike-slip and extensional focal mechanisms where the largest magnitudes are usually low to moderate (Papadopoulos -2009-, Vanucci et al., 2004). It is at the North African margin where seismicity is higher and more frequent. Tsunami sources are located offshore the Algerian 30 coasts (Fig.2) and although some submarine slides have been described offshore the Ebro delta and the Southern coast of France, recent (2003) and historical dates point out the southern provenance of the main events.*

*The 1755 Lisbon tsunami was detected at the Balearic Islands. In Majorca (the neighboring island of Minorca) it has been described by Fontsere (1918) a tsunami flooding Cala Santanyi, in the E coast of Majorca, reaching two kilometres inland and a runup of 45 m.*

*We consider very meaningful the distribution of boulder sites at the Balearic Islands (figures 1 and 3). Boulders sites in Majorca are distributed along the eastern and southern coast and the same happens in Ibiza. Only in Minorca we found boulder sites at the north coast, despite most of the boulder settings are located in the south and west coasts of the island. In figure 3 we show the perfect correspondence between the expected locations where a northern Africa generated tsunami will hit (from numerical model simulation) and the sites where the authors have found boulders accumulations. Why we haven't found large boulders at the western and northern coast of Majorca or at the northern coast of Ibiza, with similar geological features, if wind fetch is the largest in this direction? The most suitable answer is because the boulders are tsunami related.*

**4) Information on wave regime**, and tectonic setting (did your sites undergo to uplift or subsidence?) is not fully reported. In the new paragraph maritime climate, you affirm that the maximum wave height of 10 m was observed. However, where is the maximum wave height of 10 m measured? At the buoy or in the coast? Are these values in deep water, or breaking wave heights? If values are at buoy, you must consider breaking waves, which may be considerably higher than the waves in deep water (see e.g. Sunamura and Horikawa,1974). Again, data on the maximum wave heights related to historical tsunamis that hit the Minorca Island are maximum 3 m both observed and computed. If this value is in deep water, you must consider breaking wave, which may be considerably higher. Please try to compute them.

*The study sites are stable since Upper Miocene. Recent eventual uplifts are negligible for the decades or few centuries since the boulder transport was produced.*

*The maximum wave height of 10 m was measured at a buoy, according to Cañelles (2010). Later on, Jansà (2013) described a medicane causing an 11 m wave height.*

*Related to the height of the wave in deep water and the height of the wave at the coast, the observed data in the Balearic Islands is: the tsunami of 2003 had an offshore wave height of 30-40 cm (according to simulations) and reach the western part of Ibiza with a run-up of 3 m, which means a multiplying factor of x10. In the other hand, in November of 2017, a severe (the worst ever recorded) storm caused waves of up to 11 m offshore north of Minorca. These waves even decreased their height dramatically after breaking, when arriving at the coast of Menorca. We made a field campaign days after the storm and none the boulders we marked in advance (even those located at only 1 m above sea level) moved, neither new blocks appeared.*

**5) Authors show in Table 1 tsunamis observed in the Balearic Islands** and their surroundings in the last four centuries, while in Figure 2 tsunami from northern Algeria affecting Balearic Islands modelled by Roger and Hebert (2008). Both show maximum wave heights of 2 m and the May 21, 2003 tsunami was 3 m high; they were the highest tsunami waves recorded in recent years in the Balearic Islands. With reference to the studied boulders,

Authors affirm that "Our findings along the higher cliffs of the W coastline, requires tsunamis run-ups 13 m high and / or storm run-ups of 18.6 m". The observed maximum tsunami wave is 3 m high and it is 10 m less than the run-up that need to dislodge the boulders (13 m), whereas the 18.6 m storm run-up is only 7 m more than the 11 m observed! On the other hand, it is possible that storms had higher waves than you observed or that the deposition of boulders by tsunami/s occurred before your historical observation period.

*The Engel and May equation calculates the wave height needed to transport boulders located at sea level. The height at which the boulders are is not contemplated in this equation. In the next table we present the average results for joint bounded boulders (JBB) of the three sectors studied. Hs (storm wave height needed to move a boulder at sea level) is, in the three sectors, approximately four times Ht (tsunami wave height needed to move a boulder at sea level). Because Hs is approximately equal (or higher) to Hm (the maximum wave height recorded), storms of Minorca cannot move any boulder with Transport Figure>1000, not even the ones located at sea level! This point agrees with our observations in the field that the biggest storm wave ever recorded in Minorca moved none of the boulders we mark in advance. In the other hand, tsunami waves heights of less than 4 m at the northern sector or less than 2 m in the western and southern sectors can move boulders with TF>1000 at sea level. The run-up values given in the text are the sum of Hs and Alt (the average altitude of boulders), for storms run-up (Rs) and the sum of Ht and Alt for tsunami run-up (Rt).*

*According to the Engel and May equation, if Hs is four times Ht that means than a storm wave has to be four times higher than a tsunami wave for moving the same boulder. Thus, the 3 m high tsunami wave recorded in Ibiza is more energetic than the higher storm wave ever recorded (11m).*

| Sector | Hs | Ht | Alt | Rs | Rt | TF | Hm |
|--------|------|------|-------|-------|-------|-------|------|
| North | 13,79 | 3,45 | 7,81 | 21,60 | 11,26 | 8.501 | 11,0 |
| West | 6,61 | 1,66 | 11,97 | 18,58 | 13,63 | 2.404 | 8,0 |
| Southeast | 7,58 | 1,90 | 6,80 | 14,38 | 8,70 | 2.466 | 7,5 |

| | |
|------|------------------------------------------------------------------|
| Hs | *Average Storm wave height from Engel and May (2012) for JBB with TF >1000* |
| Ht | *Average Tsunami wave height from Engel and May (2012) for JBB with TF>1000* |
| Alt | *Average altitude of measured boulders* |
| Rs | *Run-up needed for storm waves (Hs+Alt)* |
| Rt | *Run-up needed for tsunami waves (Hs+Alt)* |
| TF | *Transport Figure average* |
| Hm | *Maximum wave height recorded by Cañelles (2007)* |

*In order to better understand the run-up data we introduced this table in the manuscript.*

**6) The last remark regards your dating methods**. You report different values for 14C through the text and you do not show your dating results with calibration and error. How did you date with 14C boulders 1964 AD and 1856 AD? Usually the last three centuries are uncertain in 14C dating. These boulders seem to have been recently emplaced because are among the "five of the analysed boulders showing marine fauna". Furthermore at Page 5

lines 10-14, you said "A boulder from Son Ganxo (SE of Minorca, Fig. 6) is a fragment of shoreline notch (wave-cut notch); located 2.5 m above sea level, at a distance of 18.4 m from the cliff edge, with a weight of 4.75 t. Radiocarbon dating determined an age younger than 1964 AD. Another boulder in Sant Esteve (SE of Minorca, Fig. 6) is situated about 19 meters from the waterfront and 1 m above sea level, with a weight of 43.15 t, and 14C dating determined an age younger than 1856 AD". Therefore, they are likely storm boulders. Also dating with post-depositional dissolution pans (Fig. 4b) is not useful; the range of dispersion of calculated ages makes the values overlapping. Therefore, they just show that boulders where recently deposited.

*We now show the dating results with calibration and error.*

*We agree with the referee about the significance of our results: they just remarks that boulders where dislodged and transported in recent times. This result rules out the interpretation of the boulder ridges as old coastlines. But on the other side, they don't support any interpretation as storm wave transport, neither as tsunami wave movement. Again, our interpretation as tsunami transported boulders is based not in their estimated age, but in their setting. Most boulders settings are located facing to the South, which is towards the main tsunamigenic sources of this part of the Mediterranean.*

**SPECIFIC COMMENTS (Referee 2)**

**Page 1. Lines 23:** "Some are positioned well above the maximum stand of any recorded storm wave." This is a conclusion. You must demonstrate it.

*We suppressed that sentence.*

**Page 1. Line 30:** "In fact, in many areas of the Western Mediterranean, metric size boulders have been interpreted as remnants of the tsunamis occurred in the last centuries (Pignatelly = Pignatelli et al., 2009). Yes, but other authors showed that these are storms deposited (again you should demonstrate).

*We agree there are storms deposits close to sea level in some places of the Western Mediterranean, but we also agree that there are tsunamis. So, lower sites surely are results of both processes.*

**Page 2. Lines 1-26:** Move all this part in the chapter where you describe the setting of your region and the maritime climate.

*Accepted and done.*

**Page 2 line 20.** "There are also historical records reporting a flooding event with a run-in up to 2 km inland on the east coast of Majorca (the largest of the Balearic Islands) in 1756 (Fontsere, 1918). Reformulate this sentence: There are also historical tsunami records reporting a flooding up to 2 km inland on the east coast of Majorca (Yes but in Minorca?)

*Accepted and reformulated in the manuscript.*

**Page 2, lines 7-8:** "The last seismic event recorded that affected Minorca Island was the Zemmouri (Algeria) earthquake that took place on May 21, 2003, with a magnitude of 6.9 Mw". Was it the earthquake that affected Minorca or rather Minorca was affected by the following tsunami?

*Accepted. Null reports exist for the earthquake, just the tsunami generated by the Zummari earthquake affected Minorca.*

**Page 2. Line 25.** "Thus, in the last 60 years the maximum extremal wave height detected is of 11 m at the 2001 medicane (Jansà, 2013)". See general comments, was it observed at buoy? At Page 4 line 20: in the last 50 years by a maximum wave height of 10 m (use always 60 years or 50 years).

*Yes, all the storm wave data comes from deep water buoys.*

**Page 5. Line 20** blocs = blocks    *Accepted.*

**Page 7. Line 29-30** "storm run-ups of 14 m are needed to dislodge the boulders, while tsunamis run-ups of only 8 and 13 m would explain their position".  The observed maximum tsunami wave is 3 m and it is 10 m less than the run-up that need to dislodge the boulders (13 m), whereas the 14 m storm run-up is only 3 m more than the 11 m observed!

*First of all, according to historical data (Fontsere, 1918), in the Balearic Island, the maximum tsunami run-up ever recorded was 45 m in the SE of Majorca.*

*Second, according to the Engel and May equation, to move the same block, the height of a storm wave has to be four times greater than a tsunami wave. Thus, the 3 m high tsunami wave recorded in Ibiza is more energetic than the higher storm wave ever recorded (11m) at the Balearic Islands.*

*Third, in the SE Sector, storm run-up needed is 14.38 m, tsunami run-up needed is 8.7 m and the maximum recorded wave height is 7.5 m. In the Western Sector, storm run-up needed is 18.58 m, tsunami run-up needed is 13.63 and the maximum wave height is 8 m. In the Northern Sector, storm run-up needed is 21.6 m, tsunami run-up needed is 11.26 and the maximum recorded wave height is 11 m. Thus, the difference between storm run-up needed and maximum wave height recorded is of 7.88 m (SE Sector), 10.58 m (W Sector) and 10.6 m (N Sector). In the other hand, the difference between the maximum tsunami wave recorded (3 m in Ibiza Island) and the tsunami wave height required from Engels and May equation, is of 5.7 m (SE Sector), 10.63 m (W Sector) and 8.26 m (N Sector). Thus, the differences are larger in the storm waves.*

*Forth, in the calculation of the run-up values, we add the results of hydrodynamics equations (Hs and Ht) to the altitude of the boulders (Alt) in both cases. Even, without taking into account than tsumani wave has a larger period and is more energetic (four times more energetic, according to Engel and May equation) than a storm wave, the differences between observed and required run-up will be even smaller than the mentioned before.*

**Page 8 lines 2-4:** "tsunamis run-ups 13 m high and/or storm run-ups of 18.6 m. ….and require storm run-ups of more than 21 m that are not plausible, while the height of a tsunami run-up required to position the boulders is only 9 meters." Are you sure? 13 m vs 18.6 m and 9 m vs 21 m, some calculation was wrong.

*The referee is wright. The 9 meters calculation is wrong. It has to be 11.26m.*

*Engel and May equation calculates the wave height needed to transport boulders located at sea level. The height at which the boulders are is not contemplated in this equation. According to the table presented before, the run-up values presented is the sum of (Hs + Alt) for storms waves* and *the sum of (Ht+Alt) for tsunami waves. Thus, run-up values depends on the height on which the boulder is located.*

**Page 8, lines 12-14:** "For these reasons we think than run-ups heights on Minorca would have been several times higher than tsunami wave heights. On the contrary, as they shoals, wave heights increase its run-up heights in a much lesser way and thus, it is impossible to reach the run-up values obtained from the hydrodynamic equations". Very confusing sentence. Furthermore, you must not think but demonstrate, for example computing run-up at coast using values at deep water.

*The referee is right. We must not think but demonstrate. We delete that sentence.*

**Page 8 lines 15-27** move in the study area chapter.

*Accepted.*

**Page 8 lines 28-30** "Regarding the dating of the boulders, although only two blocks with embedded marine fauna have been radiocarbon dated, such dates serve as a reference to the second dating method used. Our C14 results show than in one case a block was moved after 1720 AD, Sure? Your dating was 1856 AD. Was not it?

*Accepted. We have included all the details from C14 data.*

**Page 9, lines 25-26.** "have been dislodged and positioned by the action of tsunami waves, although some of these boulders have also been reworked by storm waves". I do not understand. Why can storms rework boulders but cannot deposit them?

*Accepted. Boulder reworking can be considered in some way a form of deposition.*

Finally, a revision of written English would be welcomed.

Figures should be quoted from the first (Figure 1) to the last (Figure 10) and not without order.

*Accepted.*

The figures and figure captions have several locality names reported in different way.

*We check that.*

Please make attention to the points and comma.

*Accepted.*

1000 in English is 1,000 and not 1.000; 14.4 m and not 14,4 etc.

*Accepted.*

[revised manuscript text omitted]

---

## Author Response (AR4)

**EDITOR COMMENTS**

Comments to the Author:
Dear Authors,

Your last corrected version of the paper "Tsunami boulders on the rocky shores of Minorca (Balearic Islands)" including the last comments of the reviewers has been satisfactorily revised for me. I have 2 little comments to be corrected in the last version of the manuscript: 1) in page 14 (line7) you have to identify if it is a significant wave height o max. wave, and (2) Figure 7 the table is not correct, you repeat table from figure 8.

*You are wright: the table is repeated!. We changed the table*
*In the new revision we clarify, that is maximum wave height.*

A last comment is regarding the tsunami 2003 in Ibiza and other locations, the amplification has been associated to the resonance induced in the harbors and in Palma for amplification effects of the bay (see e.g. Vela, J., Pérez, B., González, M., Otero, L., Olabarrieta, M., Canals, M.and Casamor, J.L.,2014. Tsunami resonance in Palma bay and harbour, Majorca Island, as induced by the 2003 Western Mediterranean earthquake. The Journal of Geology, 2014, volume 122, p. 165–182. Doi: 10.1086/675256
It is possible similar amplification effects can affect your study zone, including resonance effectes between islands.

*We agree with that statement. We have included your comment and the reference in the article.*

*Thank you very much for the comments.*
*Yours sincerely,*
*The authors.*

[revised manuscript text omitted]